# Gut microbiota metabolite tyramine ameliorates high-fat diet-induced insulin resistance via increased Ca$^{2+}$ signaling

Peng Ma ⓘ, Yao Zhang, Youjie Yin, Saifei Wang, Shuxin Chen, Xueping Liang, Zhifang Li ⓘ & Hansong Deng ⓘ ✉

## Abstract

**The gut microbiota and their metabolites are closely linked to obesity-related diseases, such as type 2 diabetes, but their causal relationship and underlying mechanisms remain largely elusive. Here, we found that dysbiosis-induced tyramine (TA) suppresses high-fat diet (HFD)-mediated insulin resistance in both Drosophila and mice. In Drosophila, HFD increases cytosolic Ca$^{2+}$ signaling in enterocytes, which, in turn, suppresses intestinal lipid levels. 16 S rRNA sequencing and metabolomics revealed that HFD leads to increased prevalence of tyrosine decarboxylase (Tdc)-expressing bacteria and resulting tyramine production. Tyramine acts on the tyramine receptor, *TyrR1*, to promote cytosolic Ca$^{2+}$ signaling and activation of the CRTC-CREB complex to transcriptionally suppress dietary lipid digestion and lipogenesis in enterocytes, while promoting mitochondrial biogenesis. Furthermore, the tyramine-induced cytosolic Ca$^{2+}$ signaling is sufficient to suppress HFD-induced obesity and insulin resistance in *Drosophila*. In mice, tyramine intake also improves glucose tolerance and insulin sensitivity under HFD. These results indicate that dysbiosis-induced tyramine suppresses insulin resistance in both flies and mice under HFD, suggesting a potential therapeutic strategy for related metabolic disorders, such as diabetes.**

**Keywords** Diet-induced Obesity; Microbiota; Insulin Resistance; Ca$^{2+}$ Signaling
**Subject Categories** Metabolism; Microbiology, Virology & Host Pathogen Interaction; Signal Transduction

## Introduction

The gut microbiota is highly dynamic and its composition is sensitive to host genetics, age, lifestyle and diet. Recent evidence suggests that obesity is associated with changes in gut microbial abundance and diversity (Hildebrandt et al, 2009; Van Hul and Cani, 2023). For instance, Diet-induced obesity (DIO) was shown to increase the percentage of Gram-positive bacteria, such as genus Clostridium, and contributes to high-fat diet-induced colonic inflammation (Wang et al, 2020). In *ob/ob* mice and humans, many studies have observed that the ratio of Firmicutes to Bacteroidetes phyla (F/B ratio) in the gut is increased (Backhed et al, 2004; Ley et al, 2005; Turnbaugh et al, 2009). The transfer of gut microbiota from *ob/ob* mice to germ-free mice resulted in a greater accumulation of adipose tissue than the transfer of gut microbiota from lean mice (Duca et al, 2014; Murphy et al, 2015), suggesting that an intestinal dysbiosis is critical for obesity development. However, a number of studies reported no or even decreased F/B ratio in obese animals and humans (Duncan et al, 2008; Tims et al, 2013; Zhang et al, 2009). Studies have also shown that probiotics from the genera Lactobacillus and Bacillus of the Firmicutes phylum have the potential to reduce obesity (Abenavoli et al, 2019; Cai et al, 2023). Overall, whether the F/B ratio change is the consequence or the cause of obesity remains unknown.

Microbiota-derived metabolites, such as, Short-chain fatty acids (SCFAs) and secondary bile acids (BAs) have been uncovered as signaling molecules linked to HFD-induced diabetes (Agus et al, 2021). For example, dietary supplementation with butyrate could prevent and treat diet-induced insulin resistance in mice (Gao et al, 2009), similar to that found in metagenomic studies of type 2 diabetes patients (Karlsson et al, 2013; Qin et al, 2012). While much remains to be characterized, these factors act on specific G protein-coupled receptors (GPCRs) or nuclear receptors to trigger the secretion of gut hormones such as glucagon-like peptide 1 (GLP-1), a major incretin that controls glucose homeostasis and improves insulin resistance (Andersen et al, 2018).

Recent evidence has also shown that the gut microbiota was actively involved in the metabolism of aromatic amino acids (AAA) (Liu et al, 2020). In addition, studies have indicated that a reduction in microbial metabolites of tryptophan is associated with insulin resistance (Virtue et al, 2019). AAA decarboxylases is responsible for decarboxylation of AAA to trace amines. For instance, *Enterococcus faecalis* in *Firmicutes* Phyla catalyze tyrosine into tyramine (TA) by Tyrosine decarboxylase (Parthasarathy et al, 2018; Pessione et al, 2009). However, the crosstalk between microbiota and host under HFD conditions, and the role of

Yangzhi Rehabilitation Hospital, Sunshine Rehabilitation Center, Frontier Science Center for Stem Cell Research, School of Life Sciences and Technology, Tongji University, 20092 Shanghai, China. ✉E-mail: hdeng@tongji.edu.cn

microbial metabolites on HFD-associated metabolic syndromes, such as insulin resistance, remains largely unknown.

The Drosophila intestine is similar in structure and function to its mammalian counterpart. Studies have also shown that diet-induced obesity (DIO) in Drosophila resembles human syndromes such as hyperlipidemia, insulin resistance, cardiac dysfunction and so on (Baenas and Wagner, 2022; Birse et al, 2010; Musselman et al, 2011). Similarly, the microbiota in Drosophila is sensitive to HFD and aging and has been shown to regulate intestinal stem cells (ISCs) proliferation (von Frieling et al, 2020) and aggression (Jia et al, 2021).

Previous studies have demonstrated that $Ca^{2+}$ signaling is a conserved negative regulator of lipid metabolism. For instance, mutants of *itpr* (the fly IP3R) or *SERCA* (a conserved ER $Ca^{2+}$ pump solely responsible for transporting $cytoCa^{2+}$ into the ER lumen), regulate lipid storage in fly fat body (Bi et al, 2014; Subramanian et al, 2013). Reducing $Ca^{2+}$ levels by loss-of-function mutations of *Orai* or *Stim*, two components of SOCE (Store Open induced Calcium Entry) channels, resulted in lipid droplets accumulation in both flies and mammals (Maus et al, 2017). However, the detailed signaling pathway and whether $Ca^{2+}$ is regulated under HFD conditions remain unknown.

Here, we found that dysbiosis-induced TA activates the $cytoCa^{2+}$ signaling pathway to suppress diet-induced obesity in Drosophila. Furthermore, TA administration is sufficient to suppress HFD-associated insulin resistance in both Drosophila and mice. The TA-mediated $Ca^{2+}$ signaling cascade mediates the crosstalk between the gut microbiota and the host, forming a negative feedback loop to regulate HFD-induced metabolic symptoms.

# Results

## cytoCa²⁺ in enterocytes was activated by HFD to reduce lipid accumulation through the Gαq/PLCβ/IP3R cascade

Enterocytes (ECs) are the main cell type in the intestine responsible for nutrition digestion and absorption of nutrients. We tested whether $cytoCa^{2+}$ participates in dietary lipid metabolism in ECs. To visualize $cytoCa^{2+}$ in ECs, a genetically encoded calcium indicator, GCaMP3, was driven by NP1Gal4 (an EC-specific driver) using the binary UAS-Gal4 system (Brand and Perrimon, 1993). Under confocal microscopy, enteric $Ca^{2+}$ is abundant around F-actin enriched microvilli, while relatively low in the cytosol (Fig. EV1A). To further quantify enteric $cytoCa^{2+}$, UAS-tdTomato-P2A-GCaMP5 (a bicistronic $Ca^{2+}$ reporter) was driven by a RU486 inducible enterocyte driver, 5966$^{GS}$Gal4 (Guo et al, 2014). The relative fluorescence ratio (GCaMP5 vs. tdTomato) was utilized to quantify $cytoCa^{2+}$ in ECs under two-photon microscopy (Fig. 1A). In contrast to what have found in ISCs (Deng et al, 2015), no oscillations of $cytoCa^{2+}$ were observed in ECs (Movie EV1). As shown in Figs. 1B,C and EV1B, $cytoCa^{2+}$ levels in ECs were robustly increased by *Serca* knockdown, whereas reduced by IP3R knockdown, suggesting that $cytoCa^{2+}$ in ECs is regulated by a conserved $Ca^{2+}$ signaling machinery.

Intriguingly, lipid levels in guts and in whole body were robustly reduced when *Serca* was knocked down in enterocytes by NP1Gal4$^{ts}$; UAS::*Serca*$^{RNAi}$ (Figs. 1D–F and EV1B). Similar results were obtained when Stim and Orai were overexpressed in ECs (Fig. EV1C,D). On the other hand, IP3R knockdown in ECs significantly increased enteric and systemic neutral lipid levels as

stained by Oil Red O(ORO) or triacylglycerols (**TAG**) level (Fig. 1D–F). Taken together, these results indicate that enteric $Ca^{2+}$ regulates intestinal and systemic lipid metabolism.

Intestinal lipid metabolism is sensitive to dietary conditions (Ko et al, 2020). We therefore tested whether enteric $Ca^{2+}$ signaling is altered by HFD. A normal diet (ND, yeast, and cornmeal) supplemented with 30% coconut oil or lard oil has been widely used to induce diet-induced obesity (DIO) (Baenas and Wagner, 2022; Birse et al, 2010). Here, we found that 4 days of HFD (supplemented with coconut oil or lard oil) was sufficient to increase gut lipid levels (Figs. 1G and EV1E,F), which was suppressed by *Serca*$^{RNAi}$ in ECs (Fig. 1H). Surprisingly, $cytoCa^{2+}$ levels in ECs were significantly higher in flies fed HFD than in those fed with normal chow (Figs. 1I and EV1G). Importantly, knockdown of IP3R or Gαq robustly suppresses HFD-induced $cytoCa^{2+}$ in ECs, while increasing gut lipid content under both ND and HFD conditions (Figs. 1J,K and EV1H,I). These results indicate that $cytoCa^{2+}$ are activated by HFD to reduce lipid accumulation in ECs.

## Tyramine induced by HFD activates enteric cytoCa²⁺ signaling

We then sought to identify the source(s) responsible for the HFD-induced increase in $cytoCa^{2+}$ levels in ECs. Intriguingly, although animals fed with a FFA mixture derived from coconut oil for 4 h show elevated enteric lipids, $cytoCa^{2+}$ in ECs is largely unchanged (Figs. 2A and EV2A). Coconut oil is enriched in middle-chain fatty acids (MCFAs) such as lauric acid (C12, 48.6%), myristic acid (C14, 19%), and palmitic acids (C16, 10%) (Sacks, 2020) (Fig. EV2B). Flies-fed diets supplemented with these MCFAs for 6 h failed to increase enteric $cytoCa^{2+}$ (Fig. EV2C). On the other hand, enteric $cytoCa^{2+}$ was increased when flies were fed for 4 h with gut lysates from HFD animals instead of those from ND animals (Fig. 2B,C). These results indicated that metabolite(s) in gut lysates contribute to HFD-activated $cytoCa^{2+}$ levels in ECs.

To identify the metabolite(s), a widely targeted metabolomic profiling was performed in gut lysates from flies fed with HFD or ND for 4 days. As shown in Figs. 2D and EV2D, 1461 metabolites were detected and 175 of them show significant changes in HFD vs ND condition by volcano plot of orthogonal projections to latent structure discriminant analysis (OPLS-DA). The supervised method, Partial Least Square Discrimination Analysis (PLS-DA), was also applied to investigate the difference of metabolites, and the R2 value of PLS-DA indicated that there was a statistically significant difference between ND and HFD samples (Fig. EV2D). KEGG analysis indicated that metabolites involved in the metabolism of arachidonic acid, α-linolenic acid, glycerolipid and glycerophospholipid metabolism were highly enriched among the differentially expressed metabolites (Fig. EV2E), suggesting that lipid metabolism is remodeled under HFD condition. Notably, metabolites involved in vitamin metabolism are also increased by HFD, which is similar to what has been shown in mammals (Hong et al, 2021; Sun et al, 2023).

The trace amine tyramine (TA) is the precursor of octopamine (OA), and both act through G protein-coupled receptors to regulate behavior, motility and metabolism in invertebrates (Roeder, 2020). Intriguingly, TA levels were significantly upregulated in gut lysates from HFD animals, whereas OA levels were largely unchanged (Figs. 2D and EV2F). Surprisingly, flies fed TA-containing chow for 1 day show a dose-dependent increase in enteric $cytoCa^{2+}$

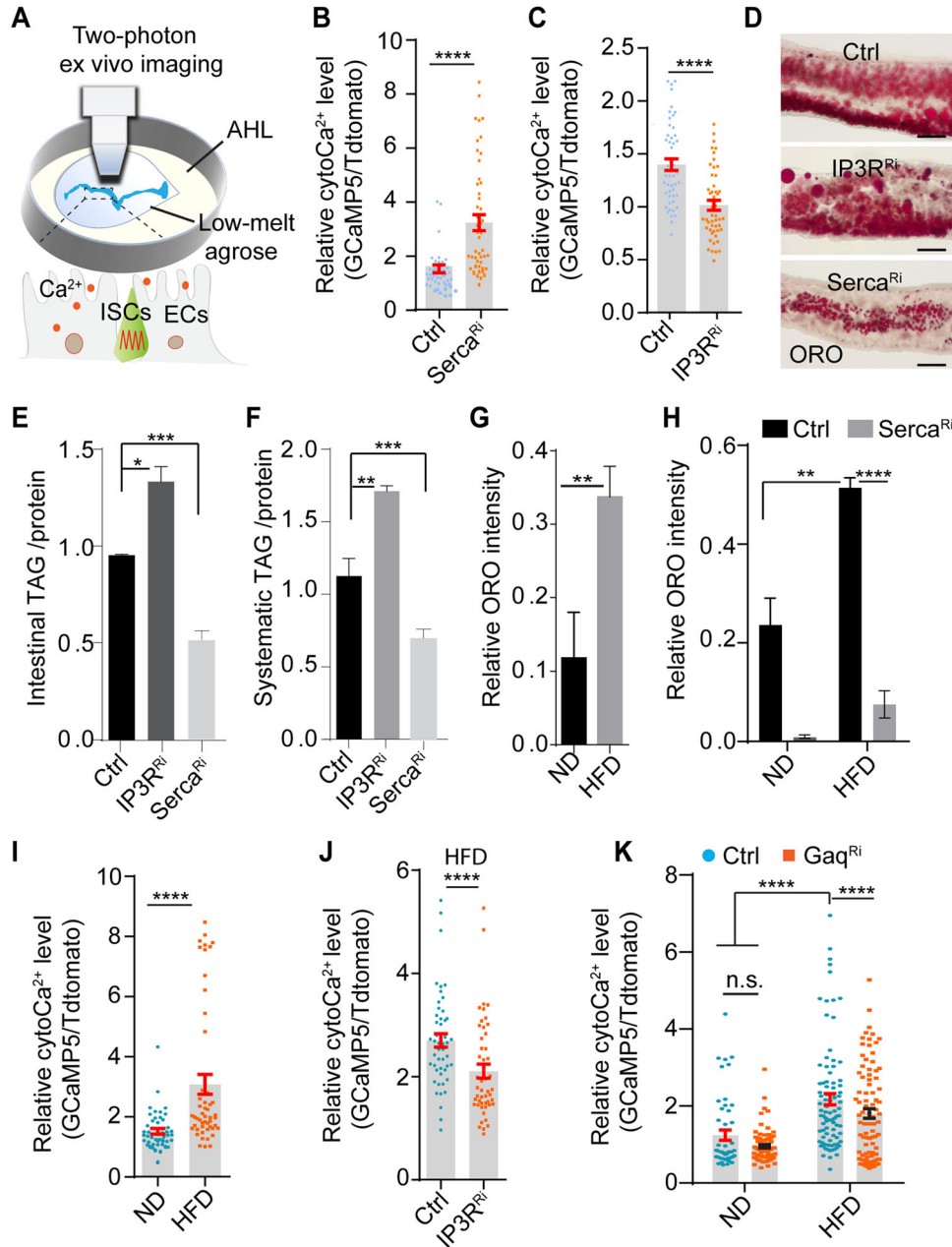

**Figure 1.  cytoCa²⁺ in enterocytes was activated by HFD to reduce lipid levels through the Gαq/PLCβ/IP3R cascade.**

(A) Schematic of two-photon live imaging setup. CytoCa²⁺ (red dots) in freshly dissected guts were incubated with low-melting agarose (1% in AHL, adult hemolymph Liquid) and imaged by two-photon microscopy. Two main cell types in the guts are shown in the bottom: ISCs (intestinal stem cells) and ECs(enterocytes). (B, C) Relative cytoCa²⁺ levels in ECs with the indicated genotypes were quantified by the fluorescence ratio of GCaMP5 vs tdTomato. Each dot represents one EC, at least 100 ECs from 6 guts were quantified. ****$P < 0.0001$. Genotypes:NP1Gal4, tubGal80$^{ts}$; UAS-*tdTomato-P2A-GCaMP5G*, UAS-*Serca$^{RNAi}$* (B) and NP1Gal4, tubGal80$^{ts}$; *UAS-tdTomato-P2A-GCaMP5G*, UAS-*IP3R$^{RNAi}$* (C). (D) Neutral lipid levels in guts were examined by Oil Red O (ORO) staining. Representative images of ORO staining in the R2 region are shown. Scale bar: 100 μm. (E, F) TAG levels in fly gut (E) and whole body (F) were quantified using a TAG kit after normalized to total protein. Triplicates were performed for each experimental set. Genotypes: NP1Gal4, tubGal80$^{ts}$; UAS-*Serca$^{RNAi}$* or NP1Gal4, tubGal80$^{ts}$; UAS-*IP3R$^{RNAi}$*. *$P < 0.05$, **$P < 0.01$, ***$P < 0.001$. (G, H) Quantification of neutral lipid levels in the R2 region after 4 days of ND (normal diet) or HFD (high-fat diet) feeding. The relative intensity of ORO staining was analyzed by Image J. At least 10 guts of each genotype were analyzed. (G) **$P < 0.01$. (H) **$P < 0.01$, ****$P < 0.001$. (I–K) Relative cytoCa²⁺ levels in ECs were monitored by two-photon live imaging after flies were fed in ND or HFD for 4 days. Each dot represents one EC, at least 100 ECs from 6 guts were quantified for each condition. (I, J) ****$P < 0.001$. (K) ****$P < 0.001$. Genotypes: NP1Gal4, tubGal80$^{ts}$; *UAS-tdTomato-P2A-GCaMP5G*, UAS-IP3R$^{RNAi}$ or NP1Gal4, tubGal80$^{ts}$; *UAS-tdTomato-P2A-GCaMP5G*, UAS-Gaq$^{RNAi}$ or NP1Gal4, tubGal80$^{ts}$; *UAS-tdTomato-P2A-GCaMP5G*, UAS-Serca$^{RNAi}$. Student *t* test for (B, C, E–G, I, J). Two-way ANOVA for (H, K). For all panels, mean ± SEM are shown. Source data are available online for this figure.

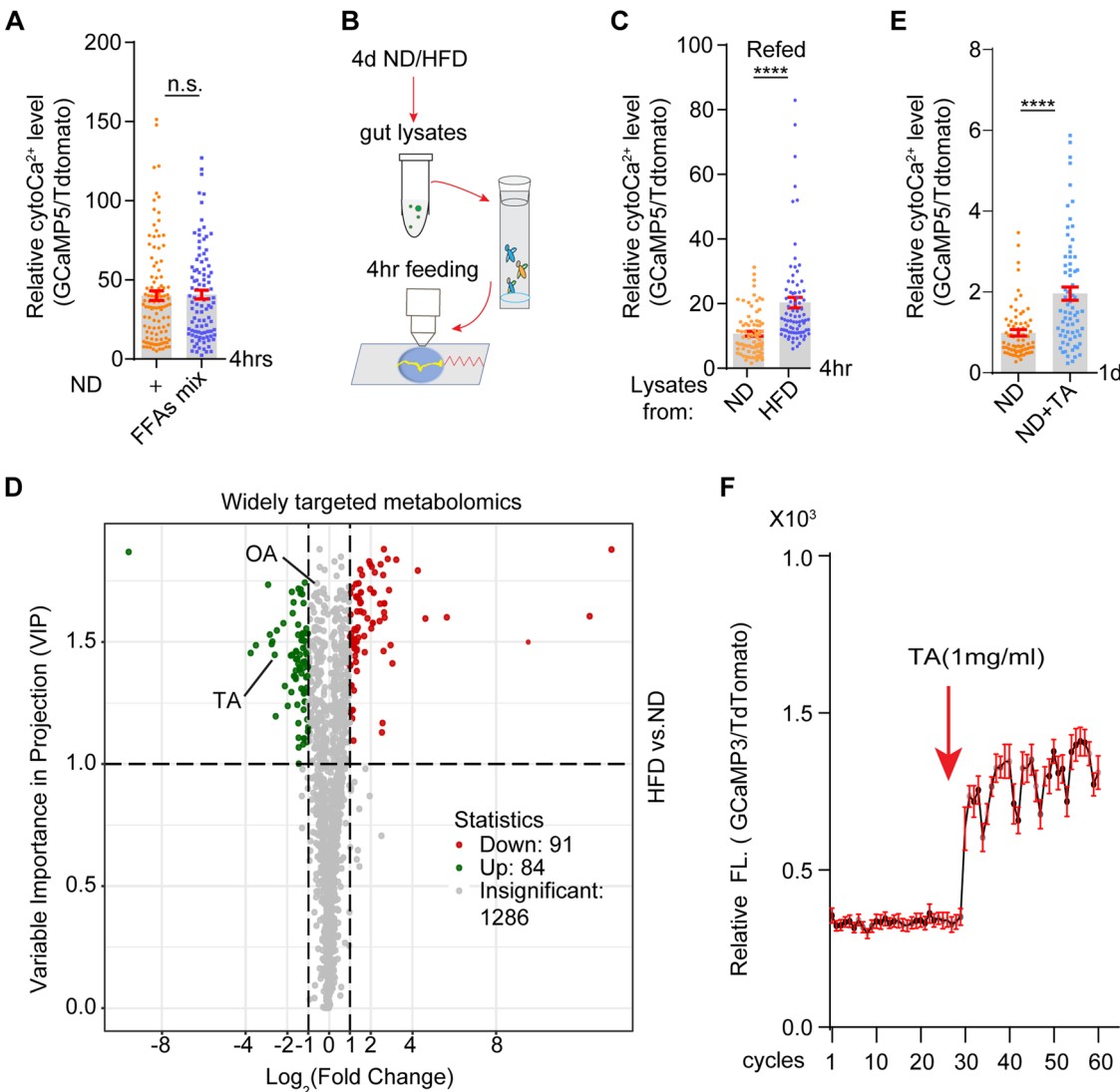

**Figure 2. Increased tyramine in the gut of HFD-fed flies activates enteric cytoCa²⁺ signaling.**

(A) CytoCa²⁺ levels in ECs were recorded after feeding for 4 h with diets supplemented with free fatty acid mixtures (FFAs mix) derived from coconut oil. Student *t* test for statistics, at least 100 ECs from 6 intestines were quantified for each condition, mean ± SEM are shown, n.s. no significance. (B) Feeding schedule. Animals were fed with the supernatant of gut lysates for 4 h. Guts were then freshly dissected and live imaged by two-photon microscopy. (C) Gut lysates from HFD-fed flies promote enteric cytoCa²⁺ after 4 h of feeding. Student *t* test for statistics, each dot represents one EC, at least 60 ECs from 6 guts were quantified. Mean ± SEM are shown. ****P < 0.0001. (D) Volcano plot of PSA-DA analysis from widely targeted metabolomics of gut lysates from flies fed ND or HFD for 4 days. Fold changes are shown on the X axis and variable importance in projection (VIP) is shown on the Y axis. Metabolite tyramine (TA) and octopamine (OA) were indicated. (E) CytoCa²⁺ was increased in ECs after flies were fed TA (1 mg/ml) containing diet for 24 h. Biological triplicates were performed. Student's *t* test for statistics, each dot represents one EC, at least 60 ECs from 6 guts were quantified. Mean ± SEM are shown. ****P < 0.0001. (F) TA exposure acutely increases enteric cytoCa²⁺ levels. Representative calcium imaging traces depicting the effect of single application of TA (1 mg/ml). In all, 10 s/cycle, 60 cycles were recorded. At least 20 ECs from 4 guts were quantified. Mean ± SEM are shown. Source data are available online for this figure.

compared to those fed control chow (Figs. 2E and EV2G). Meanwhile, TA exposure acutely increases enteric cytoCa²⁺ in freshly dissected guts (Fig. 2F; Movie EV2), and this cytoCa²⁺ increasing effect can last at least for an hour (Fig. EV2H).

## Gram-positive bacteria increased by HFD are responsible for TA production

We then sought to identify the source of TA under HFD conditions. The Drosophila genome encodes two *Tdc* genes: *Tdc2* is exclusively

expressed in neurons, and Tdc1, which is expressed non-neuronally (Cole et al, 2005), was not readily detected in the adult fly gut (Buchon et al, 2013). Consistently, Ca²⁺ levels in ECs were largely unaltered by gut-specific knockdown of *Tdc1* (Fig. EV3A), and systematic inhibition of *Tdc2*, whose neuronal expression was comparable under ND and HFD conditions (Fig. EV3B), failed to suppress HFD-induced lipid accumulation in guts (Fig. EV3C). TA was also fermented from tyrosine by tyrosine decarboxylase (TDC)-expressing bacteria in the diet (Schoeler and Caesar, 2019). The *Tdc* gene was encoded in the genome of several Gram-positive bacterial species belonging to the

genera *Lactobacillus* and *Enterococcus*. We first tested whether HFD regulates the fly gut microbiome. As shown in Figs. 3A and EV3D, the bacterial load in gut lysates of flies fed with HFD for 4 days was significantly increased, and which was further increased after 15 days of HFD feeding compared to those in the ND.

Interestingly, feeding the animals with antibiotic cocktail (ampicillin, vancomycin, neomycin, metronidazole, and tetracycline) significantly reduces enteric cytoCa$^{2+}$ in gut lysates of flies fed with HFD for 4 days (Fig. 3B). Ca$^{2+}$ levels in ECs are also significantly reduced in germ-free flies under HFD conditions (Fig. EV3E). Furthermore, germ-free or antibiotic-treated flies can further increase intestinal lipid content under HFD conditions (Figs. 3C and EV3F,G). These results suggest that the gut microbiome is involved in HFD-mediated cytoCa$^{2+}$ regulation.

The gut bacterial composition in flies fed in ND versus HFD was then compared by 16 S amplicon sequencing. As shown in Fig. EV4A,B, the abundance and diversity of bacterial species are significantly increased in the HFD condition. Gram-positive bacterial species, especially those belonging to the genus *Lactobacillus* were enriched by HFD (Figs. 3D and EV4C). Intriguingly, feeding flies with wild-type but not *tdc*-deleted strain of *Lactobacillus Brevis* (ATCC8287) for 24 h was sufficient to increase cytoCa$^{2+}$ in ECs (Figs. 3E and EV4D). However, flies fed for 24 h with *Lactobacillus Plantarum (BNCC187903)*, another Lactobacillus species that does not have the *Tdc* gene in its genome failed to increase enteric cytoCa$^{2+}$ (Figs. 3F and EV4E). On the other hand, flies fed with *Enterococcus Faecalis (ATCC 29212)*, another *Tdc*-expressing Gram-positive microbe that is not a commensal for fly gut, were able to significantly increase enteric cytoCa$^{2+}$ level (Figs. 3F and EV4E). Meanwhile, thin-layer chromatography (TLC) showed a robust increase in TA levels in the culture medium of *L. Brevis* and *E. Faecalis*, but not *L. Plantarum* (Fig. 3G), which was suppressed by nicotinic acid (NA, 0.5%, 12 h) (Fig. EV4F), a potent TDC inhibitor with low toxicity (Kang et al, 2018; Zhang and Ni, 2014). Moreover, HFD or *L. Brevis*-induced Ca$^{2+}$ elevation in ECs was significantly reduced by NA supplementation (Figs. 3H and EV4G). These results indicated that Gram-positive bacteria were enriched under HFD conditions, and that TA produced by these Tdc-expressing bacteria was responsible for Ca$^{2+}$ induction in ECs under HFD conditions.

## Tyramine produced by Gram-positive bacteria activates enteric Ca$^{2+}$ signaling through the tyramine receptor *TyrR1*

We then investigated how tyramine activates enteric Ca$^{2+}$ signaling. Knockdown of *Gαq* or *PLCβ* is sufficient to suppress TA or HFD-mediated cytoCa$^{2+}$ induction in ECs (Figs. 4A and EV5A), suggesting the involvement of GPCR(s) in TA or HFD-mediated Ca$^{2+}$ upregulation. There are three tyramine receptors in the Drosophila genome (Zhang and Blumenthal, 2017). *TyrR2* (CG16766) and *Oct-TyrR* receptor (CG7485) are activated by TA with a weak selectivity for TA over OA (Bayliss et al, 2013), whereas *TyrR1* (CG7431) is a highly selective receptor for TA. All of them can trigger Ca$^{2+}$ release when activated (Zhang and Blumenthal, 2017). RT-qPCR experiments indicated that *TyrR1* instead of *TyrR2* or *Oct-TyrR* receptor was highly expressed in the intestine (Fig. EV5B). Knockdown of *TyrR1* instead of *TyrR2* or *Oct-TyrR* can significantly block TA-induced cytoCa$^{2+}$ levels by ex vivo

imaging (Figs. 4B and EV5C). Meanwhile, *TyrR1* knockdown in ECs but not in enteroendocrine cells was sufficient to increase intestinal lipid levels and suppress HFD-induced Ca$^{2+}$ elevation (Figs. 4C and EV5D,E).

CREB is a conserved transcription factor that is regulated by cytoCa$^{2+}$ to control various biological functions, such as ISC proliferation (Deng et al, 2015) and memory formation(Silva et al, 1998). Indeed, CREB activity (indicated by nuclear staining of phosphor-CREB$^{S133}$, and the luciferase-based activity reporter 5xCRE-LUC) was significantly increased in enterocytes after 4 days of HFD feeding (Fig. 4D,E). CREB has been shown to be phosphorylated by calcium-calmodulin kinases (CaMKs) (Heist and Schulman, 1998). Indeed, *CaMKII* knockdown in ECs substantially blocked HFD-associated CREB activation, as indicated by the mean intensity and percentage of p-CREB positive ECs in the R2 region (Fig. 4F,G). Flies with *Serca* knockdown in ECs, or fed a TA-containing diet, or fed wild-type but not *Tdc*$^{-/-}$ *L. Brevis*, have elevated levels of p-CREB in the intestine (Fig. 4H–J). Moreover, HFD-induced CREB activation was suppressed by *Gαq* knockdown (Fig. 4K), suggesting that HFD activates CREB through the TA/TyrR1/Gαq mediated Ca$^{2+}$ cascade.

*CREB*$^{Δ36}$ is a deletion allele of CREB (Fig. EV5F). As shown in Fig. 4L, *CREB*$^{Δ36}$ mutants accumulated more intestinal lipids than controls under both ND and HFD conditions. Intriguingly, overexpression of CRTC (an evolutionarily conserved coactivator of CREB) (Guo et al, 2014; Hirano et al, 2013) in ECs with NP1Gal4$^{ts}$ was sufficient to reduce HFD or *Gαq*$^{RNAi}$-induced intestinal lipid accumulation (Figs. 4M,N and EV5G). These results indicate that the CRTC/CREB cascade activated by cytoCa$^{2+}$ in ECs suppresses intestinal and whole-body lipid levels.

## *Magro* expression was transcriptionally repressed by the cytoCa$^{2+}$/CRTC/CREB cascade to inhibit dietary lipid digestion

We then examined how enteric lipid metabolism is regulated by the cytoCa$^{2+}$/CRTC/CREB cascade. Transcriptional profiling experiments were performed to compare differentially expressed genes (DEGs) regulated by overexpression of CRTC or IP3R in the guts. Venn diagram analysis revealed that nearly 57% (240/416) DEGs are shared by CRTC$^{OE}$ and IP3R$^{OE}$ in guts, (Appendix Fig. S1A). Consistent with our previous findings (Yin et al, 2022), genes involved in proteostasis, calcium homeostasis were among these DEGs. KEGG pathway enrichment analysis indicated that these shared DEGs are also enriched in lipid and carbon metabolism pathways (Appendix Fig. S1B). Intriguingly, *magro*, a well-characterized lipase responsible for the digestion of dietary lipids (Ma et al, 2021; Sieber and Thummel, 2012), was significantly reduced by both CRTC$^{OE}$ and IP3R$^{OE}$ (Fig. 5A), as confirmed by RT-qPCR experiments (Fig. 5B). Bioinformatic analysis indicated that nearly 73% (303/416) of DEGs containing at least one canonical CRE motif in their promoter region (Appendix Fig. S1C), suggesting that these genes are unlikely transcriptionally regulated by CREB. *Magro* expression was also significantly reduced by genetically increasing enteric cytoCa$^{2+}$ levels (Stim$^{OE}$, Orai$^{OE}$ or Serca$^{RNAi}$) or by HFD feeding (Fig. 5C; Appendix Fig. S1D).

Bioinformatic analysis revealed that there are two conserved CRE motifs in the upstream 2 kb region of *magro* promoter (Fig. 5D). ChIP-qPCR analysis showed that the fragment

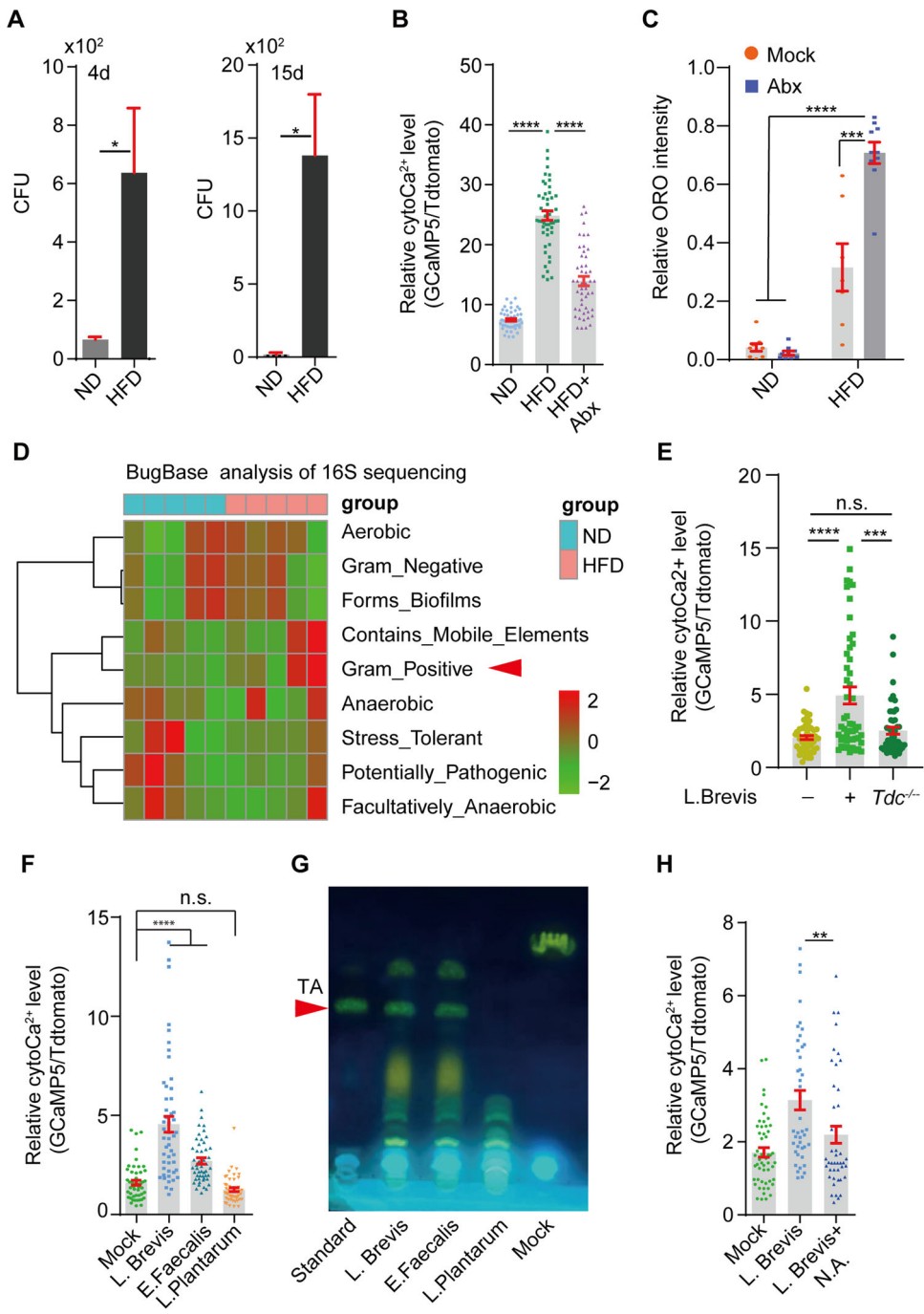

**Figure 3. Gram-positive bacteria increased by HFD is responsible for tyramine production.**

(**A**) CFU (colony-forming unit) assay quantified the bacteria in gut lysates after 4 or 15 days of HFD feeding. Triplicates were performed for each condition. *$P < 0.05$.
(**B**) Antibiotics (Abx) cocktail reduces HFD-mediated cytoCa²⁺ elevation in ECs. ****$P < 0.001$. In total, 80–100 ECs from 6 guts were quantified for each condition.
(**C**) Antibiotics cocktail further increases HFD-associated neutral lipid levels in the guts. ORO intensities in gut R2 region were quantified by Image J. ***$P < 0.001$,
****$P < 0.0001$, $n = 6$. (**D**) BugBase analysis of 16 S amplicon sequencing showed that Gram-positive bacteria were significantly increased by HFD. Arrowhead denotes
Gram-positive bacteria. (**E**) Germ-free flies fed with wild-type instead of *Tdc*-deleted strain (*Tdc⁻/⁻*) of *L. Brevis* promote cytoCa²⁺ in ECs. ****$P < 0.0001$, ***$P < 0.001$,
n.s. no significance. (**F**) Flies fed with *L. Brevis* or *E. Faecalis* instead of *L. Plantarum* for 24 h is sufficient to increase cytoCa²⁺ in ECs. ****$P < 0.0001$. (**G**) Thin-layer
chromatography (TLC) analysis indicated that tyramine (TA) levels were robustly increased in the culture supernatant of *L. Brevis* or *E. Faecalis*, but not in *L. Plantarum*. The
arrowhead indicates the band of TA. (**H**) Nicotinic acid (N.A.), a potent TDC inhibitor, robustly reduces L. Brevis-mediated cytoCa²⁺ induction in ECs. Each dot represents
one EC. **$P < 0.01$. Genotypes: NP1Gal4, tubGal80ᵗˢ; UAS-tdTomato-P2A-GCaMP5G. Student *t* test for (**A, B, E, H**) and two-way ANOVA for (**C, F**). For all panels,
mean ± SEM are shown. For Ca²⁺ live imaging in (**E, F, H**), 80–100 ECs from 6 guts were quantified for each condition. Source data are available online for this figure.

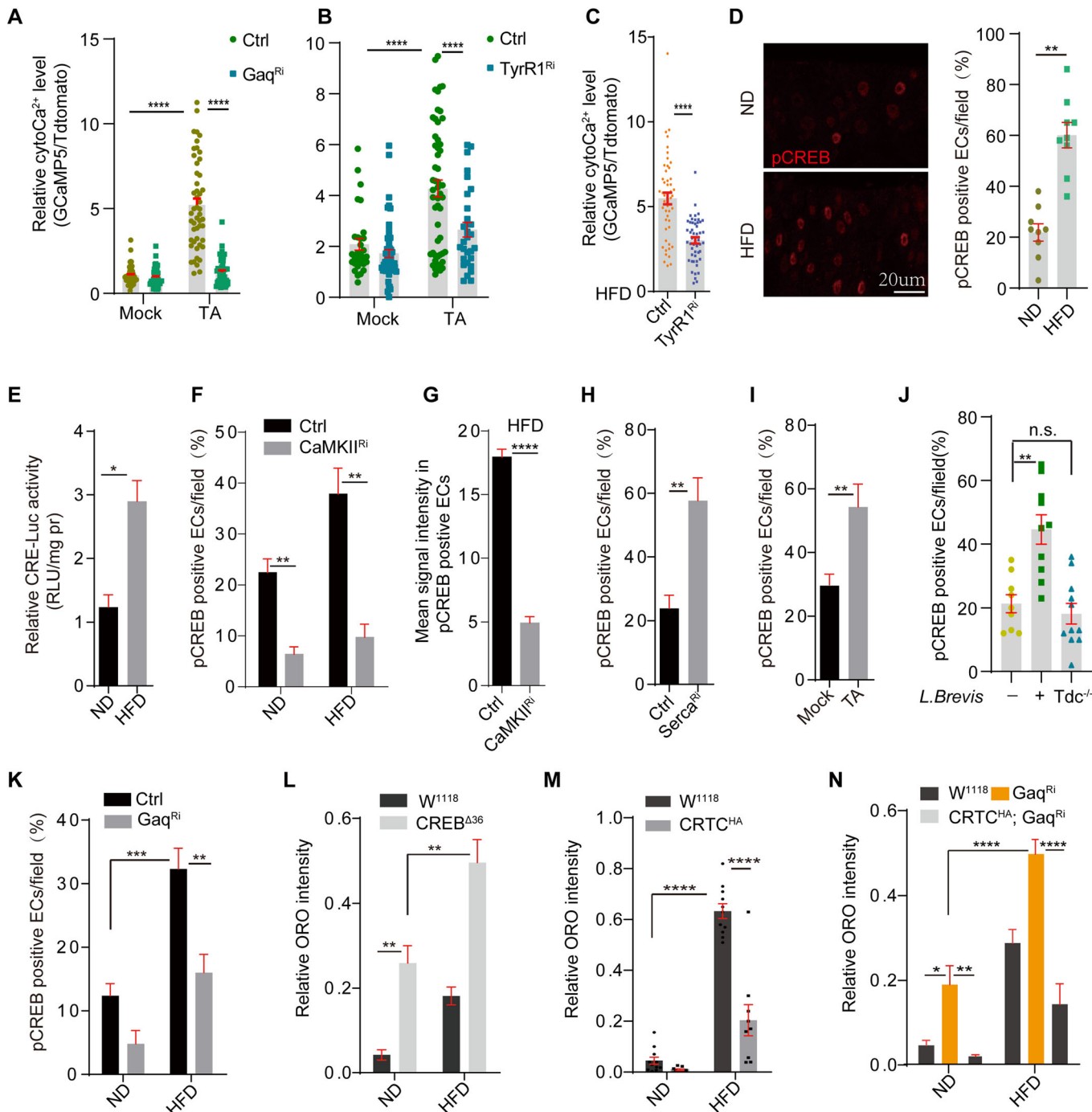

containing CRE-1 (-907 to -915) was significantly enriched in gut lysates by antibody against CREB compared to IgG (Fig. 5D), suggesting that CREB can directly bind to the promoter of *magro*.

Consistent with its role on dietary lipid digestion, *magro* inhibition significantly suppressed intestinal lipids under both HFD and ND feeding conditions (Fig. 5E). TAG levels in fecal samples were increased by gut-specific overexpression of CRTC (Fig. 5F; Appendix Fig. S1E), suggesting that lipase activity of Magro was perturbed. Consistently, lipase activity in the gut lumen was markedly reduced under HFD or by CRTC overexpression

(Fig. 5G). Whereas supplementation with pancreatin, a lipase mixture (Ma et al, 2021), can significantly restore gut lipid levels reduced by CRTC overexpression (Fig. 5H; Appendix Fig. S1F).

## The Ca²⁺/CRTC/CREB cascade promotes mitochondrial biogenesis while inhibiting lipogenesis in enterocytes

In enterocytes, FFAs are resynthesized into TAGs, which form micelles and are transported to other tissues via micelles (Phan and Tso, 2001). Intriguingly, the expression of *DGATs* (CG1942 and

**Figure 4.  Tyramine produced by Gram-positive bacteria activates enteric Ca²⁺ signaling through the tyramine receptor *tyrR1*.**

(A, B) Gut-specific knockdown of *Gaq* (A) or *tyrR1* (B) in ECs is sufficient to suppress TA-mediated cytoCa²⁺ induction. ****$P < 0.0001$. Overall, 60–80 ECs from five intestines were quantified for each condition. (C) HFD-induced cytoCa²⁺ levels in ECs were also inhibited by *TyrR1*ᴿᴺᴬⁱ. ****$P < 0.0001$. Genotypes: NP1Gal4, tubGal80ᵗˢ; UAS-tdTomato-P2A-GCaMP5G, UAS-*Gaq*ᴿᴺᴬⁱ for (A), and NP1Gal4, tubGal80ᵗˢ; UAS-tdTomato-P2A-GCaMP5G, UAS-*TyrR1*ᴿᴺᴬⁱ for (B, C). (D) Representative images showing that CREB activities (indicated by anti-phosphor-CREB signals in red) in ECs are increased after HFD feeding. Quantifications are shown on the right. **$P < 0.01$, ****$P < 0.0001$. Scale bar: 20 μm. (E) CREB activities (indicated by luciferase activities of CRE-LUC normalized to protein level) in guts are increased after HFD feeding. Intestinal lysates from 8 intestines were analyzed. Triplicates were performed. *$P < 0.05$. (F, G) *CaMKII* knockdown in ECs suppresses HFD-induced CREB activation. The number of p-CREB-positive ECs (F) and mean signal intensity of p-CREB in ECs (G) were quantified. **$P < 0.01$. Genotypes: NP1Gal4, tubGal80ᵗˢ; UAS-*CaMKII*ᴿᴺᴬⁱ. (H) Knockdown of *Serca* specifically in the intestine promotes p-CREB activities in ECs. **$P < 0.01$, $n = 10$. (I) TA-containing diet (1 mg/ml, 1d) are sufficient to increase p-CREB-positive ECs in the intestine. For (H, I), CREB activities were quantified by anti-p-CREB staining. **$P < 0.01$. (J) CREB activity in ECs was quantified after flies were fed with *L. brevis*. *Tdc⁻/⁻*: *tdc* deletion strain of *L. brevis*. $n = 8$–12 for each condition. **$P < 0.01$, n.s: no significance. (K) HFD-induced CREB activation (anti-p-CREB staining signal) was suppressed by *Gaq*ᴿᴺᴬⁱ. **$P < 0.01$, ***$P < 0.001$. Genotype: NP1Gal4, tubGal80ᵗˢ; UAS-*Gaq*ᴿᴺᴬⁱ. (L) More lipids accumulated in the intestines of *CREB* mutants (CREBᐩᴰ³⁶) under both ND and HFD conditions. Three independent experiments were performed. $n = 8$ for each condition, **$P < 0.01$. (M) Gut-specific CRTCᴼᴱ is sufficient to suppress HFD-induced neutral lipid accumulation. Three independent experiments were performed. $n = 8$ for each condition. ****$P < 0.0001$. Genotype: NP1Gal4, tubGal80ᵗˢ; UAS-CRTCᴴᴬ. (N) CRTCᴼᴱ is sufficient to suppress *Gaq*ᴿᴺᴬⁱ-induced lipid accumulation under both ND and HFD conditions. Three independent experiments were performed. $n > 8$ for each condition. *$P < 0.05$, **$P < 0.01$, ****$P < 0.0001$. (L–N) The intensity of ORO staining in the R2 region was quantified. Genotypes: NP1Gal4, tubGal80ᵗˢ; UAS-CRTCᴴᴬ or NP1Gal4, tubGal80ᵗˢ; UAS-CRTCᴴᴬ, UAS-*Gaq*ᴿᴺᴬⁱ. Student *t* test for (A–J), and two-way ANOVA for (K–N). For all panels, mean ± SEM are shown. For Ca²⁺ live imaging in (A–C), around 80 ECs from 6 guts were quantified for each condition. For quantification of p-CREB intensity in (D, F–K), ~100 ECs from 6 to 7 intestines were analyzed. Source data are available online for this figure.

CG1946), essential genes involved in lipogenesis from FFAs, were both downregulated by CRTC overexpression in ECs (Fig. 6A). FFAs can also be utilized by mitochondria as a respiratory substrate for ß-oxidation. Among the DEGs genes regulated by CRTCᴴᴬ and IP3Rᴼᴱ, genes involved in mitochondrial replication, such as TFAM and mtSSB (Picca and Lezza, 2015), are significantly upregulated (Fig. 6A). Consistently, mitochondrial DNA copies (indicated by qPCR for mtDNA16S), mitochondrial mass (indicated by mitochondrial-targeted GFP under spinSR microscopy) and western blot against ATP5A (a mitochondrial ATP synthase subunit) in intestine, all indicated that mitochondrial biogenesis was increased by CRTCᴼᴱ (Fig. 6B,C; Appendix Fig. S2A,B). Western blot against ATP5A and transmission electron microscopy (TEM) images also showed that the mitochondrial content in HFD ECs was significantly higher than in ND ECs (Appendix Fig. S2C,D). Intestinal ATP levels and ROS production (as shown by DHE staining) were also significantly increased in CRTCᴼᴱ or HFD conditions (Appendix Fig. S2E,F), suggesting that β-oxidation was activated by the HFD-mediated Ca²⁺ signaling cascade.

To further track the fate of FFAs in enterocytes, a pulse-chase experiment was performed using BODIPY-C₁₂, a fluorescent analog of FFA (Kolahi et al, 2016). After 4 h feeding on BODIPY-C₁₂ supplemented diet, flies were refed on normal diet for another 2 h (Fig. 6D). In controls, BODIPY-C₁₂ signals colocalized with lipid droplets (stained by LipidTOX a fluorescent neutral lipid dye) in both enterocytes and abdominal adipocytes (Fig. 6E,F). However, although still colocalized with LipidTOX in enterocytes, BODIPY-C₁₂ signals were barely detectable in abdominal adipocytes in ECs from gut-specific CRTC over-expressing flies or flies fed with TA (1 mg/ml, 24 h), despite similar levels of BODIPY-C₁₂ were taken up (Fig. 6F; Appendix Fig. S2G–I). Quantification also showed that more LipidTOX and BODIPY-C₁₂ double-positive puncta were observed in ECs from CRTC overexpressing flies or flies fed with TA, suggesting local consumption of fatty acids (Fig. 6G,H; Appendix Fig. S2J). Indeed, TAG levels in adipocytes were significantly reduced by TA feeding in both ND and HFD conditions (Fig. 6I). Taken together, these results indicate that cytoCa²⁺ in ECs suppresses dietary lipid digestion and enteric lipogenesis, while promoting mitochondrial biogenesis (Fig. 6J).

## TA-mediated Ca2+ signaling cascade suppresses HFD-associated insulin resistance

HFD feeding is associated with development of insulin resistance in both mammals and Drosophila (Lillioja et al, 1993; Yaribeygi et al, 2021). Akt-SPARK is a phase-separation based genetic reporter of Akt activity (Li et al, 2022). As expected, Akt-SPARK droplets in ECs were dramatically increased when ND-fed intestines were exposed ex vivo to insulin (5 μM, 20 min) (Fig. 7A,B). However, the droplets from ECs of chronically HFD-fed animals are weaker and fewer even after insulin incubation (Fig. 7A,B). tGPH is a genetic reporter of PI3K, and plasma membrane bound tGPH correlates well with insulin activity (Britton et al, 2002). As shown in Appendix Fig. S3A, chronic HFD feeding significantly decreases the plasma membrane localization of tGPH in ECs. Interestingly, both insulin insensitivity and lipid accumulation in ECs under chronic HFD conditions were significantly reversed by TA supplementation (1 mg/ml) or gut-specific CRTC overexpression (Fig. 7C,D; Appendix Fig. S3A,B).

The fat body is important for the regulation of lipid and glucose homeostasis in Drosophila (Padmanabha and Baker, 2014). However, insulin signaling, as indicated by membrane tethering tGPH, was largely unchanged in HFD (Appendix Fig. S3C). Mitochondrial biogenesis in the fat body was also comparable in the HFD compared to the ND, even after TA feeding (Appendix Fig. S3D). These results indicate that HFD feeding causes insulin resistance in the intestine, and further elevation of enteric Ca²⁺-mediated CRTC/CREB activity is sufficient to alleviate the diabetic phenotype (Fig. 7E).

## TA alleviates HFD-induced insulin resistance in mice

In mammals, excessive TA is considered toxic due to the release of adrenaline, resulting in elevated systolic blood pressure (SBP) or migraines, especially in patients on monoamine oxidase inhibitor (MAOI) therapy (Andersen et al, 2019). However, studies have also shown that prolonged ingestion of TA in drinking water (0.04%, around tenfold of spontaneous dietary intake) does not cause adverse cardiovascular effects in mice (Carpene et al, 2016), while its role in HFD-induced insulin resistance remains elusive. Oral

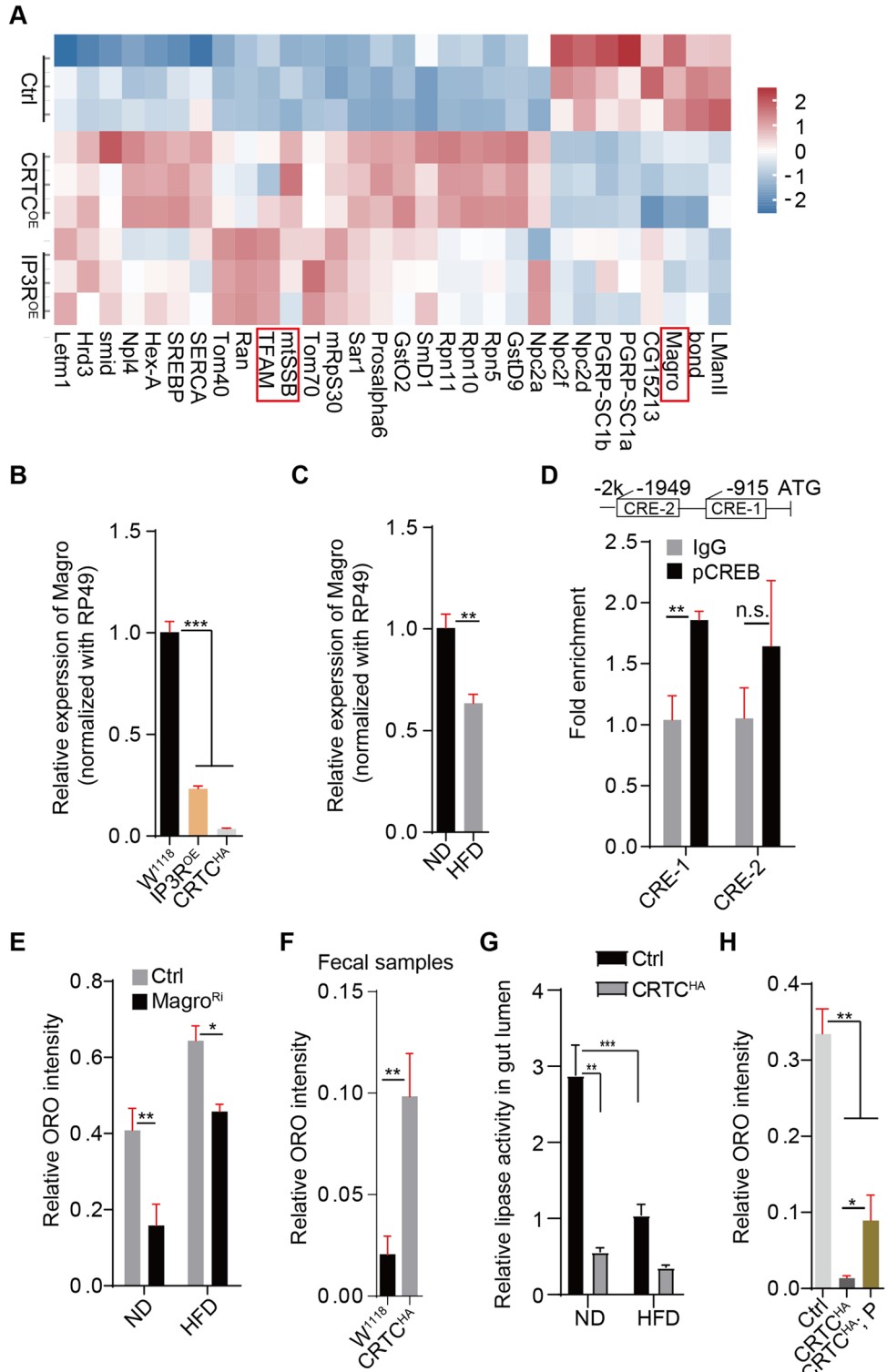

glucose tolerance test (OGTT) showed that fasting blood glucose levels and the area under the curve (AUC) of blood glucose response were significantly elevated in HFD groups (Fig. 8A,B), indicating that the HFD groups have impaired glucose tolerance. Intriguingly, TA ingestion (0.04% in drinking water, three weeks) significantly reduce blood glucose levels in HFD conditions,

although food intake is largely comparable between these groups (Fig. 8A,B; Appendix Fig. S4A). The insulin tolerance test (ITT) was utilized to evaluate insulin sensitivity. The AUC also shows a significant difference between HFD and HFD + TA groups. Notably, the decrease of blood glucose levels in ND groups were significantly restored by TA (Fig. 8C,D), suggesting that TA helps

**Figure 5.** The expression of *magro* was transcriptionally suppressed by the Ca²⁺/CRTC/CREB cascade to inhibit dietary lipid digestion.

(A) Heatmap of RNAseq analysis showing the similarity of genes regulated by CRTC^OE and IP3R^OE in the guts. (B, C) qRT-PCR analysis of *magro* expression in the intestine. Triplicates were performed for statistical purposes. ***$P < 0.001$ for (B). **$P < 0.01$ for (C). (D) Top: Bioinformatic analysis indicated that *magro* contains two conserved CRE motifs in the promoter region (upstream 2 kb region). The ChIP-qPCR assay showed that the CRE-1-containing fragment was significantly enriched by the p-CREB antibody compared to IgG. Biological triplicates performed for each condition, $t$ test for statistics, **$P < 0.01$, n.s: no significance. (E) Enteric lipids were reduced when *magro* was inhibited in the guts. Genotype: NP1Ga4^ts > UAS-*magro*^RNAi. Three independent experiments were performed. $N = 8$–10 for each condition. Student $t$ test for statistics, mean ± SEM are shown. *$P < 0.05$, **$P < 0.01$. (F) Lipid levels in fecal samples were increased by CRTC overexpression. Quantifications are shown. Three independent experiments were performed. $n = 6$ for each condition. Student $t$ test for all panels, mean ± SEM are shown. **$P < 0.01$. (G) Lipase activity in the gut lumen was measured under the indicated conditions. Mean ± SEM for three independent replicates. Student $t$ test for all panels. **$P < 0.01$, ***$P < 0.001$. (H) Pancreatin can partially reduce CRTC^OE-induced neutral lipid reduction in the guts. Pancreatin (P, 5 mg/ml) was added to the diet. Three independent experiments were performed. $n > 8$ for each condition. Student $t$ test for all panels, mean ± SEM are shown. *$P < 0.05$, **$P < 0.01$. Genotype: NP1Gal4, tubGal80^ts or NP1Gal4, tubGal80^ts; UAS-*CRTC*^HA. Source data are available online for this figure.

## Discussion

The gut microbiota is now considered as a dynamic ecosystem that plays an important role on host metabolism. The host–microbiota interactions is very complicated due to the high degree of crosstalk both within and between kingdoms. Metabolites, as the functional output of combined host-microbe interactions, play a key role in energy metabolism and mediating microbiota–gut communication (Van Treuren and Dodd, 2020). However, it remains largely unknown whether changes in microbial metabolites are the cause or effect of host metabolism in response to environmental cues.

Tyramine is commonly found in fermented, ripened foods, such as cheese, where microbes with decarboxylase enzymes convert tyrosine to tyramine (Andersen et al, 2019). TA in invertebrates is an important neurotransmitters and regulates similar biological processes with its vertebrate counterpart, adrenaline and noradrenaline (Roeder, 2020). They are not only closely related in structure, but also share physiological effect, such as the famous "fight-or-flight" response (Berger et al, 2019; Roeder, 2020). Here we found that TA was increased in HFD-fed flies due to the enrichment of Gram-positive bacteria. Although TA is not the only metabolite that changes on HFD in our metabolomic analysis, feeding germ-free flies with *Tdc*⁻/⁻ mutant L. Brevis failed to increase Ca²⁺ signaling in ECs. Moreover, HFD-mediated Ca²⁺ signaling in ECs was significantly suppressed by nicotinic acid, a Tdc inhibitor. These results indicated that TA-mediated Ca²⁺ signaling cascade in ECs is critical to reduce lipid uptake, increase lipid utilization, and ameliorate insulin resistance in HFD-fed flies (Fig. 8E). Upregulation of TA-mediated Ca²⁺ signaling in ECs is therefore an adaptive mechanism for the host to counteract diet-induced insulin resistance.

Similar to what we found in fly guts, several probiotic Lactobacillus strains are also accumulated in HFD mice, which are rich sources of tyramine (Ley et al, 2005) (Broadley et al, 2009; Straub et al, 1995).

In HFD-fed mice, glucose intolerance and insulin resistance were also significantly improved by TA ingestion, while obese phenotypes remain largely unchanged (Fig. 8E). It might explain why HFD-associated dysbiosis contributes to obesity in mice, even with elevated TA level. In the liver, the CRTC/CREB transcriptional complex promotes gluconeogenesis (Herzig et al, 2001; Wang et al, 2009). During fasting, this process is activated for tissue demand of glucose, which is consistent with elevated blood glucose level observed in ND + TA group by ITT assay (Fig. 8C,D). However, as we observed here in HFD flies, the CRTC/CREB complex was overactivated in diabetic patient, which may contribute to the elevation of fasting glucose levels in these patients (Benchoula et al, 2021). On the other hand, in small intestine, the CRTC2/CREB-dependent transcriptional pathway was shown to be a critical regulator for maturation and secretion of GLP-1 from the L cells (Lee et al, 2018). Overall, the CREB cascade has a complex effect on glucose homeostasis in mammals, and depends on the regulatory network in different organs, cell types and upstream signals. However, the gut microbiome and its regulatory mechanisms are much more complicated in mice, and whether TA regulation of glucose homeostasis is dependent on Ca²⁺/CREB cascade all await further study in the future.

In humans, excessive tyramine is considered toxic due to the release of adrenaline, resulting in hypertension which was enhanced by MAOIs, known as the "cheese effect". In rats, the doses of TA generally used to induce a threshold pressor response vary between 14 and 20 mg/kg in solution (Fankhauser et al, 1994; Humphrey et al, 2001), but rise to 67 mg/kg in solid food. In TA-drinking mice, even a dose of 185 mg/kg (0.14%) failed to increase systolic blood pressure (Carpene et al, 2016), suggesting that its pressor effect is dependent on the duration and route of administration.

To conclude, we found that dysbiosis-induced tyramine suppressed insulin resistance in both flies and mice under HFD conditions. Tyramine supplementation, at the appropriate dose and route, is therefore a potential therapeutic target for the treatment of HFD-associated symptoms, such as insulin resistance.

## Methods

### Fly food and husbandry

*W*^1118(BL3605), UAS::IP3R^OE(BL30742), UAS::PLCβRNAi(BL92505), UAS-IP3R^RNAi(BL25937), CREB^Δ36(BL79018), 20XUAS-GCaMP3 (BL32235), tGPH(BL8164), UAS::Gaq^RNAi(BL30735), UAS-Life-actRFP(BL58713), 5xCRE-LUC (BL79016), 5xCRE-mCherry(BL79020),

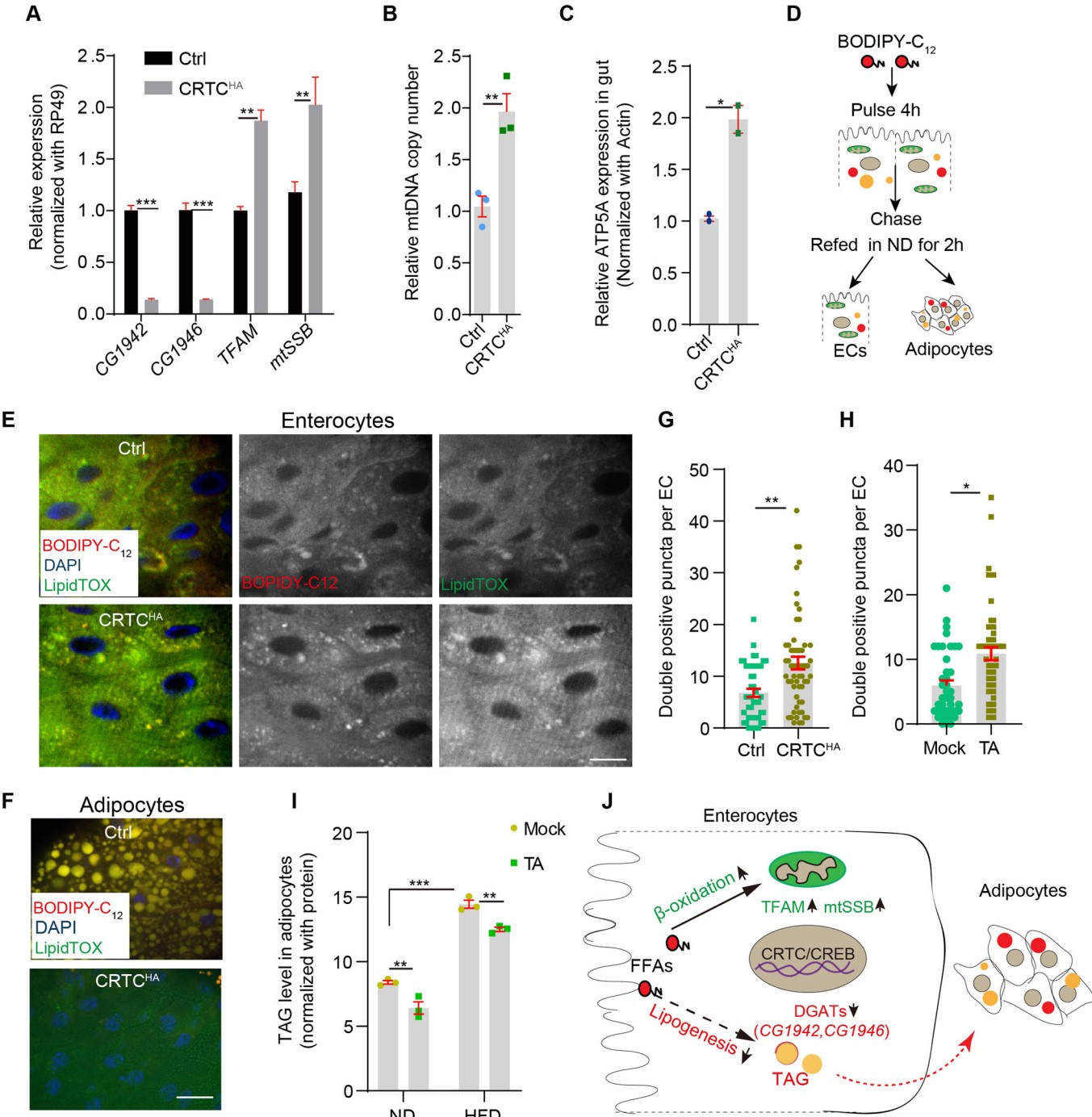

**Figure 6. The Ca²⁺/CRTC/CREB cascade promotes mitochondrial biogenesis and inhibits lipogenesis in enterocytes.**

(A) qRT-PCR analysis of gene expression in the intestine. Triplicates were performed, *t* test for statistics, mean ± SEM are shown. \*\**P* < 0.01, \*\*\**P* < 0.001.
(B) Mitochondrial DNA copy number is quantified by qPCR relative to nuclear genome (RPL32). Triplicates were performed, *t* test for statistics, mean ± SEM are shown. \*\**P* < 0.01. (C) Quantification of ATP5A expression in guts by western blot. Biological triplicates were performed, *t* test for statistics, mean ± SEM are shown, \**P* < 0.05. (D) Schematic of BODIPY-C₁₂ pulse-chase experiments. Flies were fed with BODIPY-C₁₂ containing AHL for 4 h, and then refed in ND for a further 2 h before examining its fate in different tissues. (E, F) Distribution of BODIPY-C₁₂ in ECs (E) and adipocytes (F) was examined by Spinning Disk Confocal Super Resolution (spinSR) microscopy. LipidTOX stains neutral lipids, and DAPI counterstains DNA. Genotypes: NP1Gal4, tubGal80ᵗˢ; for top panel and NP1Gal4, tubGal80ᵗˢ; UAS-CRTCᴴᴬ for bottom panel. Scale bars: 10 μm. (G, H) Quantification of BODIPY-C₁₂ and LipidTOX double-positive puncta in ECs in guts with CRTCᴼᴱ (G) or after TA feeding (1 mg/ml, 24 h) (H). Around 60 ECs from 5 guts were quantified for each condition. Mean ± SEM are shown, Student *t* test for statistics, \*\**P* < 0.01 for (G), \**P* < 0.05 for (H). (I) TAG levels in fat body were examined with a TAG kit. Means ± SEM from biological triplicates. Two-way ANOVA analysis for statistics, \*\**P* < 0.01, \*\*\**P* < 0.001. (J) Schematics: metabolic flux of free fatty acids (FFAs) in ECs. FFAs in ECs are either oxidized in mitochondria or undergo lipogenesis and transport to distal tissues, such as abdominal fat bodies. The CRTC/CREB complex transcriptionally suppresses lipogenesis by inhibiting expression of DGATs (*CG1942* and *CG1946*), while promoting mitochondrial biogenesis (*mtSSB* and *TFAM*). The CRTC/CREB complex hence facilitates the local utilization of FFAs instead of being transported to distal tissues. Source data are available online for this figure.

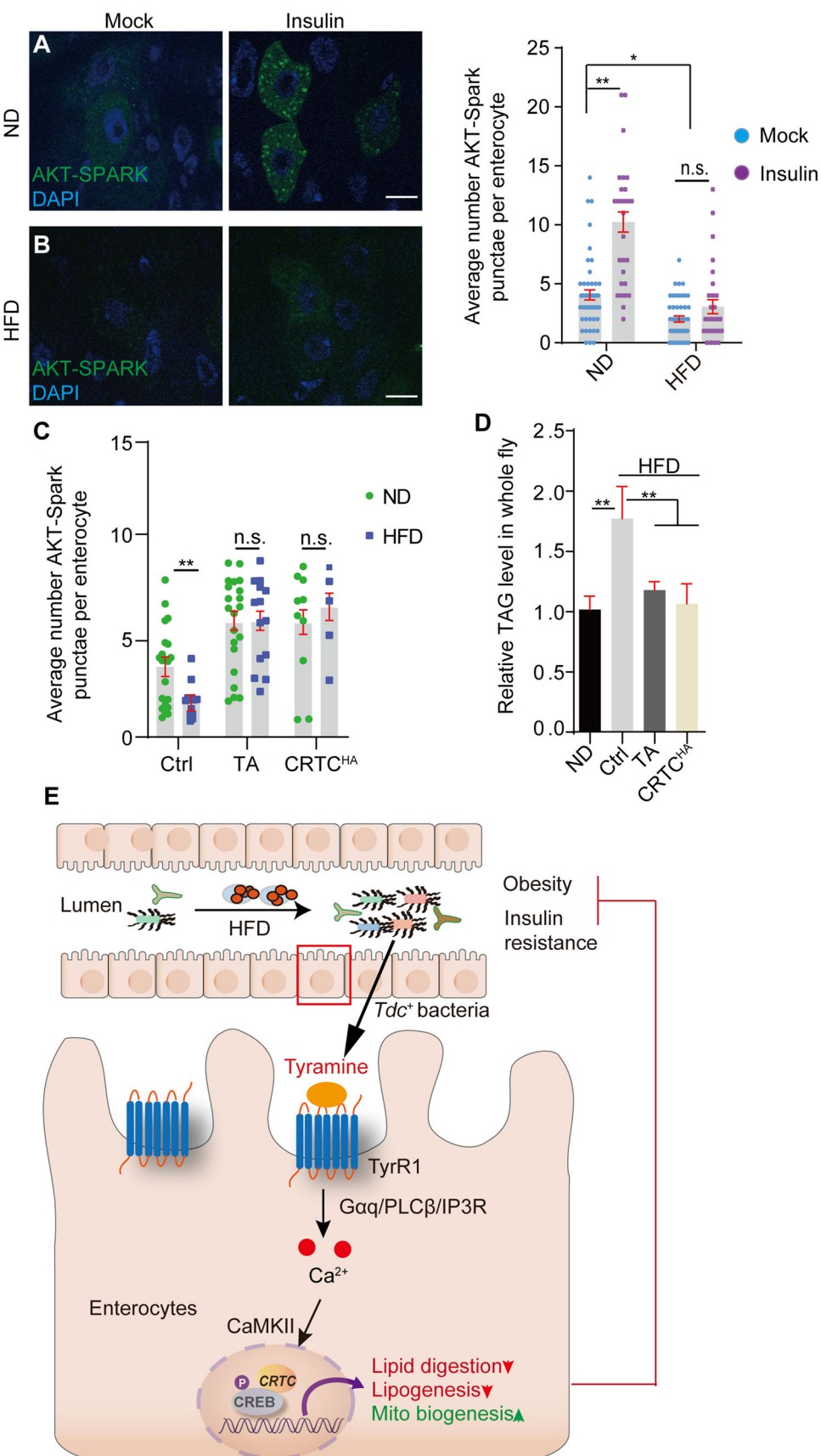

◄ **Figure 7. Tyramine-mediated Ca²⁺ signaling cascade suppresses HFD-mediated insulin resistance.**

(A, B) AKT activity in ECs in response to insulin stimulation (5 μM, 20 min). AKT activity was monitored by AKT-SPARK puncta. Quantifications are shown on the right. Approximately 50 ECs from 6 guts of each condition were analyzed. $t$ test for statistics, mean ± SEM are shown. *$P < 0.05$, **$P < 0.01$, n.s: no significance. Scale bars: 20 μm. (C) AKT-SPARK puncta in ECs of HFD-fed flies were significantly restored by TA supplementation (1 mg/ml) or gut-specific CRTC overexpression. About 50 ECs from 6 guts of each condition were analyzed. $t$ test for statistics, mean ± SEM are shown. **$P < 0.01$, n.s: no significance. (D) TA supplementation (1 mg/ml) or gut-specific CRTC overexpression suppresses systemic hyperlipidemia of HFD flies. TAG level of six animals were analyzed for each group, biological triplicates were performed. **$P < 0.01$. Genotypes for (A–D) NP1Gal4, tubGal80ᵗˢ; UAS-AKT-SPARK, UAS-*CRTC*ᴴᴬ or NP1Gal4, tubGal80ᵗˢ; UAS-AKT-SPARK. For all panels, mean ± SEM is shown. $t$ test for (A, B), and two-way ANOVA analysis for (C, D). (E) Model figure: In Drosophila, chronic HFD feeding causes dysbiosis and accumulation of Gram-positive bacteria in the gut. *Tdc* (tyrosine decarboxylase) expressing bacteria would then catalyze the production of Tyramine (TA) from tyrosine (Tyr). TA activates cytoCa²⁺ signaling in enterocytes through the GPCR/Gαq/PLCβ/IP3R cascade. The cytoCa²⁺-mediated CRTC/CREB transcriptional complex systematically reprograms lipid metabolism and suppresses insulin resistance. Source data are available online for this figure.

*UAS::mitoGFP(BL8442)* from Bloomington Drosophila Stock Center *UAS::Serca*ᴿᴺᴬⁱ(V107446), *UAS-TyrR1*ᴿᴺᴬⁱ(V2857), *UAS-TyrRII*ᴿᴺᴬⁱ(V51387), *UAS-Tdc2*ᴿᴺᴬⁱ(V330541), *UAS-Oct-TyrR*ᴿᴺᴬⁱ(V26876), *UAS-magro*ᴿᴺᴬⁱ(V109706) from Vienna Drosophila Resource Center,*UAS::CaMKII*ᴿᴺᴬⁱ(THU4064) from Tsinghua Fly Center. *UAS-tdTomato-P2A-GCaMP5G* from R.W. Daniels, *UAS-Stim*ᴼᴱ, *UAS-Orai*ᴼᴱ from G. Hasan, *NP1-GAL4 from D. Ferrandon, UAS-CRTC*ᴴᴬ *from Y. Hiran, 5966GS from H. Jasper, UAS-Akt-SPARK* from H. Huang.

Flies were cultured on normal diet (ND, which is a standard yeast/molasses-based fly food (recipe: 10 L H₂O, 138 g agar, 220 g molasses, 750 g malt extract, 180 dry yeast, 800 g corn flour, 100 g soy flour, 62.5 ml propionic acid, 20 g methyl 4-hydroxybenzoate, and 72 ml ethanol)) or on high-fat diet (HFD, which is ND supplemented with 30% virgin coconut oil or lard oil). Flies were maintained at 25 °C, 60% humidity with a 12 h: 12 h light–dark cycle. HFD are greasy and flies are easily sticky to the food, especially at temperatures above 25 °C. For HFD experiments, fly diet were changed daily and extra oil in the food were absorbed by Kimwipe paper.

For antibiotic diet, 200 μl of antibiotic cocktail (100 μg/ml ampicillin, 50 μg/ml vancomycin, 100 μg/ml neomycin, 100 μg/ml metronidazole, and 50 μg/ml tetracyclin) was added to the surface of ND or HFD and then air dried before use. For pancreatin-containing food, 5 mg/ml was added in ND or HFD.

## Luciferase assays

CRE luciferase activity was measured using the Steady-Glo luciferase assay kit (Promega Cat# E2510) according to the manufacturer's instructions. In brief, lysates from whole flies ($n = 5$) or guts from 15 animals were obtained by homogenized in 100 μL GLO lysis buffer. After centrifugation at $12,000 \times g$ for 10 min, 30 μL of supernatant were aliquoted in triplicates in 96-well plates. Three independent samples from each condition were analyzed. After incubation in the dark for 1 min, luminescence was measured using a microplate reader (Synergy HTX, BioTek, Winooski, Vermont, USA). Luminescence values were then normalized with protein concentrations, which were determined with BSA as a standard using a bicinchoninic acid (BCA) protein assay kit (Yeasen Biotechnology, Cat#20201ES76) according to the manufacturer's instructions.

## Ex vivo Drosophila intestine imaging setup

The imaging setup was based on our previous publications (Deng et al, 2015; Morris et al, 2020) with minor modifications.

Flies were dissected in Adult Hemolymph-like Saline (AHLS) culture medium containing 2 mM CaCl₂, 5 mM KCl, 5 mM HEPES, 8.2 mM MgCl₂, 108 mM NaCl, 4 mM NaHCO₃, 1 mM NaH₂PO₄, 5 mM trehalose and 10 mM sucrose. Guts were immediately transferred to 35 mm Nunc™ Glass Bottom Dishes (Thermo Scientific™, 150682), embedded in 1% low-melting agarose (in AHLS) and immersed in AHLS. For all experiments except Fig. 2G, guts were fully embedded. In Fig. 2F and supplemental Fig. 2g, the guts were only embedded in their most anterior and posterior regions. This "exposed" setup allowed for acute addition of tyramine during live imaging.

## Image acquisition

For cytoCa²⁺ live imaging, image stacks consisted of five optical sections with 1-μm Z-steps and Z-stacks were recorded every 20 s for 5 min. These were obtained using a two-photon laser microscope customized for in vivo imaging (FVMPE-RS confocal microscope, Olympus; tsunami, Spectra-Physics Inc.) with a ×25 water objective lens (N.A. 1.05). All of the fluorescence signals with 975-nm wavelengths were detected using the Non-Descanned Detector.

For cytoCa²⁺ imaging, z-stack images were converted to mean intensity projections, and automatic image stabilization, background subtraction, image segmentation, and ROI detection were performed using Image J. Mean intensity traces were then measured over time in each ROI.

## Immunostaining and microscopy

Immunostaining was performed based on previous publications (Deng et al, 2015). In brief, tissues were first dissected in 1× PBS and then fixed in 4% formaldehyde for 45 min at room temperature. After washing in washing buffer (PBS, 0.5% BSA, 0.1% Triton X-100) for 1 h, tissues were incubated with primary antibodies and secondary antibodies diluted in washing buffer. Samples were then mounted and imaged using the Zeiss AxioImager M2 with the apotome system. Images were then processed with ZEN and Image J software. Antibodies used in the studies: rabbit anti-p-CREB (Cat#9198 Cell Signaling Technology), 1:300, DNA was counterstained by Hochest22338.

For LipidTOX™ staining, fixed guts were briefly washed in washing buffer for 3× 5 min, then directly stained with LipidTOX™ solution (1:500) diluted in washing buffer. LipidTOX™ kit was purchased from Thermo Fisher (Cat # H34476).

For cross-sectioning, the guts were first fixed in 4% formaldehyde. After washing, the guts were sectioned using fine surgical blades (Saferlife, SL04-049A), the sectioned tissue were mounted

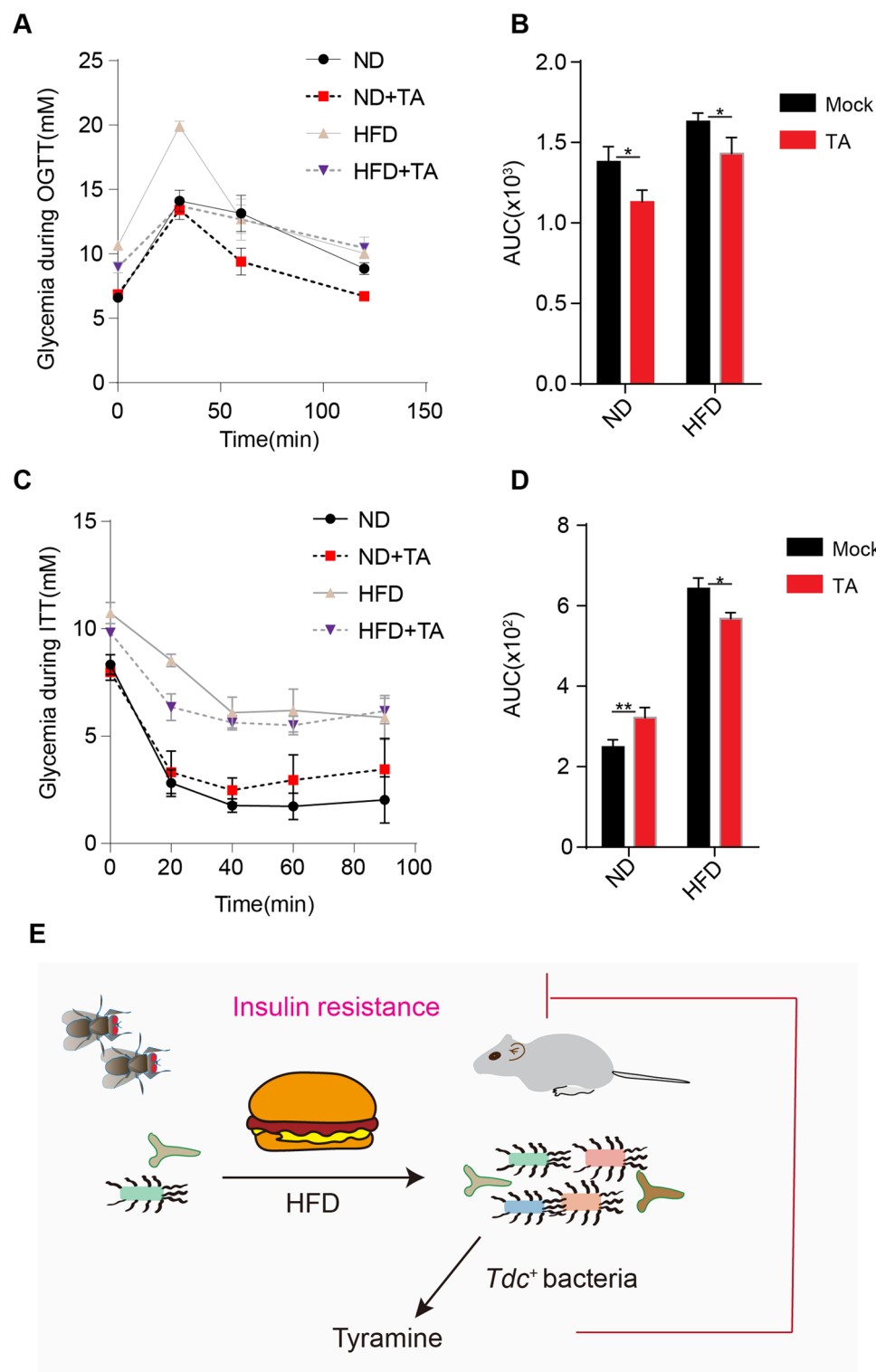

**Figure 8. TA alleviates HFD-induced insulin resistance in mice.**

(**A–D**) Oral glucose tolerance test (OGTT) and insulin tolerance test (ITT) in high-fat diet (HFD)-fed WT mice treated with Tyramine. Twelve-week-old male WT mice were fed HFD plus TA (0.04% in drinking water) for 3 weeks. Blood samples were collected from tail vein for glucose measurement at indicated time points after glucose administration (**A, B**) or insulin injection (**C, D**). The area under curve (AUC) shown in (**B, D**). Data shown represent the mean ± SEM $n = 6$ per group. Two-way ANOVA test for statistics. *$P < 0.05$, **$P < 0.01$, n.s.: no significance. (**E**) Chronic HFD feeding causes dysbiosis in both flies and mice. Dysbiosis-induced tyramine forms a conserved negative feedback loop to suppress HFD-induced insulin resistance. Source data are available online for this figure.

and imaged by Zeiss AxioImager M2 with the apotome system or by confocal super-resolution microscopy from Olympus (SpinSR). The images were acquired and analyzed using the cellSens imaging software. For TEM, the guts was dissected under a dissection microscope, fixed with 2.5% glutaraldehyde in 0.05 M phosphate buffer (pH 7.4) for 1.5 h at 4 °C, and washed three times with 0.05 M phosphate buffer at 4 °C, post-fixed with 1.0% $OsO_4$ for 1 h at 4 °C, washed with phosphate buffer, dehydrated in ethanol, and embedded in Spurr resin. Ultra-thin sections of the embedded guts were double-stained with uranyl acetate and lead citrate and examined with a JEOL 100 C transmission electron microscope.

## Quantitative real-time PCR

RNA was isolated from the guts of 15 flies using TriZol reagent (Life Technologies). Approximately 20 μg of RNA was then reverse transcribed using 5× All-In-One RT Mastermix with Accurt Kit (#G592, Applied Biological Materials) according to the manufacturer's instructions. Real-time PCR was performed using a CFX96TM Real-Time System (Bio-Rad Laboratories). Transcript levels were normalized to Rp49. Primers included: RP49 (forward: TCCTACCAGCTTCAAGATGAC, reverse: CACGTTGTGCACCA GGAACT),

Magro (forward: AGCACGGAACCTCCTTCATCTTCA, reverse: TAGTGGATGCCCTGGTTGCTAGA),

mtSSB (forward: CGCATGCTGAATCCTCTGTT, reverse: GGTCACCGTGTTGACAGTTT),

*TFAM* (forward: GGAGATGGAGCTCTACAGGAAA, reverse: GGCTCAGGTGGATCGATAAGA),

CG1942 (forward: CAACTACACCTTTGGCTTCCTC, reverse: GACCTGTCCATGCACTTTATCC),

CG1946 (forward: CGCTGGAAACTCGTCGAAATAG, reverse: ATGACGTAACCAGGGTCTGAA),

TyrR1 (forward: CTGGCTGGGTTGGTTCAATAG, reverse: AAGAAGCGTTTGCAGGTGAG)

TyrR2 (forward: GCAGAGCCATCACAATCATCAC, reverse: TTAGGCCGCCCATCACAATA)

Oct-TA receptor (forward: CCCTGGTTCTCTCGGTCATTAT, reverse: CGCCAGCGAAACTATGAAGAAG).

## ORO staining

For fly experiments, guts were dissected in PBS and fixed in 4% formaldehyde/PBS for 20 min. The guts were then washed twice in PBS and incubated with fresh Oil Red O solution (6 mL of 0.5% Oil Red O in isopropanol and 4 mL of demineralized water, passed by a 0.45-μm filter) for 30 min, without agitation. The guts were then rinsed twice in distilled water and mounted in mounting medium (60% glycerol in ddH2O). Quantification of the ORO signal in the R2-R3 region of the midguts was performed as described in (Buchon et al, 2013; Luis et al, 2016), using Image J. Briefly, images were converted to 8 bit and a constant minimum threshold was applied to the entire image for the red channel. The area above the threshold was then measured in the anterior midgut, to estimate the amount of local neutral lipids.

## Measurement of triglyceride (TAG) in flies

Triglycerides were measured using a commercially available kit (A110-1-1, Nanjing Jiancheng Bioengineering Institute). For fly experiments, flies with heads removed or guts dissected in fresh PBS were snap-frozen in liquid nitrogen. Frozen samples were then homogenized in PBS for measurement of triglycerides and free fatty acids at 510 nm according to the manufacturer's instructions. Total protein was measured using the Bradford assay for normalization.

## Quantification of fluorescence signal intensity

All images were analyzed by confocal microscopy using an LSM 700 (Carl Zeiss). To quantify fluorescence signal intensity, a midgut R2 region was imaged at ×20 magnification. Fluorescence signal intensity was quantified using ZEN image software in the region of interest (ROI) across all genotypes. Data were collected from at least 20 midguts per genotype, and the fluorescence signal intensity of each gut was represented as a dot.

## Measurement of lipase activity in fly gut lumen

Gut luminal lipase activity was measured using a lipase test kit (A054-2-1, Nanjing Jiancheng Bioengineering Institute). Briefly, fly guts were freshly dissected in 1× PBS on ice. The samples were cut into pieces with scissors and incubated at 4 °C for 30 min with gentle shaking, followed by centrifugation at 12,000 × *g* for 5 min at 4 °C. Lipase activity in the supernatant was measured at 580 nm according to the manufacturer's instructions. Total protein was measured using the Bradford assay for normalization.

## Akt-SPARK Imaging and quantification

Akt-SPARK signals in the guts were imaged by FluoView FVMPE-RS Multiphoton Microscope(Olympus, Japan), after excitation at 695 nm wavelength, and detected via the Non-Descanned Detector. For quantitative analysis of the SPARK signal, images were processed in Image J. The sum of droplets pixel fluorescence intensity and the cells pixel intensity were scored using the Analyze Particle function in Image J, as previously described (Li et al, 2022).

For insulin sensitivity in ECs, freshly dissected guts were incubated with 5 μM insulin for 15 min. The guts were then fixed in the fixative solutions, and imaged after washing and mounting.

## Bacterial strains and culture conditions

*L. Brevis* (ATCC8287) was purchased from Biofeng, Shanghai, China, *L. Plantarum* (BNCC187903) was purchased from BNCC, Beijing, China, and *E. Faecalis* (ATCC 29212) was purchased from Huankai, Guangzhou, China. *L. Brevis* was cultured in a MRS (HB0384-1, Hopebio, Qingdao, China) medium at 30 °C, 220 rpm. Tyramine content was measured by TLC (Thin-Layer Chromograph). Bacteria at OD 50 was centrifuged, and bacteria pellets were suspended in 5% sucrose and fed to the animals for 24 h. For the nicotinic acid (N.A) experiment, bacteria were cultured with MR media plus N.A.(final concentration 0.5% in 1 × PBS) at 30 °C, 220 rpm for 24 h. After centrifugation, the TA level in the supernatant was measured by TLC, and bacteria pellets were fed to the flies for $cytoCa^{2+}$ recording.

To test whether *Tdc* was present in the bacterial genome, bacterial DNA was extracted using the UltraClean Microbial DNA Isolation Kit (Qiagene, Germany). PCR primers: *Tdc* (forward: GCAGATGGTTCCTTGGCTAATC, reverse: GCACCTTCCAACT TCCCATATC).

### *Tdc* deletion strain of *Lactobacillus brevis* ATCC8287

Tdc deletion was generated based on the homologous recombination-based strategy. Lactobacillus Brevis ATCC8287 genomic DNA was extracted and used as the template to amplify the upstream and downstream segments of the *Tdc* gene. Then, the fragments were ligated by overlapping PCR and subcloned into pGhost4. The recombinant plasmid was transformed into MC1061 competent cells, plated on LB plates containing erythromycin, and cultured in inverted form overnight at 37 °C. After verified by sequencing, the plasmid was transformed into *L. brevis* by electroporation. In brief, the L. brevis competent cells were mixed with 1 µg plasmids and then were chilled for 5 min on ice. The mixture was then transferred to a prechilled electroporation cuvette (0.1 cm; Bio-Rad) to electroporate at 2.5 kV. Then, 1 ml prechilled MRS liquid medium supplemented was immediately added to the cuvette. The mixture was transferred to a sterile tube to culture for 3 h at 37 °C and then spread on MRS agar plates supplemented with 4 µg/ml erythromycin.

Primer sequence

PGhost4-Δtdc-L-F: C G G G G T A C C tcgggagata ttcaaaggtt gacactt

PGhost4-Δtdc-L-R: CGGCCGCTGTTGCTGCCGTAATTCTTTTTCgtcacttacc tcctgcag at ttaata

PGhost4-Δtdc-R-F: GAAAAAGAATTACGGCAGCAACAGCG GCCGgccaatcaa tccagtttga ctga

PGhost4-Δtdc-R-R: C T A G T C T A G A g actgtgatca accaacaaaa tcc.

### TM widely targeted metabolomics of fly gut samples

Fifty gut samples from flies fed with ND or HFD were freshly collected, frozen in liquid nitrogen and kept at −80 °C until use. Samples were then extracted and analyzed by UPLC-MS/MS using a "TM" Widely-Targeted Metabolomics combines Untargeted metabolomics and Targeted metabolomics to achieve the perfect combination of high-resolution, wide-coverage, high-sensitivity, and precise quantification (Metware Biotechnology Co., Ltd. Wuhan, China). Metabolite identification and quantification were based on the database MWDB from Metware. The supervised multivariate method of partial least squares-discriminant analysis (PLS-DA) was used to maximize the metabolome differences between sample pair. The relative importance of each metabolite to the PLS-DA model was checked using the variable importance in projection (VIP). Metabolites with a VIP ≥ 1 and a fold change ≥2 or a fold change ≤0.5 were considered as differentially accumulated metabolites for group discrimination in the ND vs. HFD.

### Tyramine measurement by thin-layer chromatography (TLC)

Tyramine was semi-quantified by thin-layer chromography (TLC) as described previously (Garcia-Moruno et al, 2005). Briefly, tyramine was converted to its fluorescent dansyl derivative by adding one V of the filtered supernatant to one volume of 250 mM disodium phosphate (pH 9.0), 0.1 V of 4 N sodium hydroxide solution, and 1 V of dansyl chloride solution (5 mg/ml of dansyl chloride in acetone). After thorough mixing, the solutions were incubated at 55 °C for 1 h in the dark. The samples were then cooled and kept at 4 °C until use. Five microliters of each supernatant was spotted on a silica TLC plate (Aluminum Sheets Silica gel 60 F254, Merck, Darmstadt, Germany). The dansylated compounds were separated using a solvent mixture of chloroform: triethylamine (4:1 v/v). The fluorescent dansyl derivative spots were visualized with the aid of a transilluminator with a suitable UV-light source (312 nm).

### 16 S rDNA sequencing

To extract commensal genomic DNA from the guts, the flies were dipped into 70% ethanol for about 1 min to kill bacteria on the fly cuticle and were then dissected in 1× sterile PBS. The fly crops were removed while leaving the whole midguts intact to avoid leakage. Each sample of 10 female guts was processed using the UltraClean Microbial DNA Isolation Kit (Qiagene, Germany). The DNA was used as templates for limited cycle PCR with primers targeting V3/ V4 regions (forward 5'-CCTACGGGNGGCWGCAG-3' and reverse 5'- GACTACHVGGGTATCTAATCC-3') to get the 16 S metagenomic sequencing library. Reaction conditions: 94 °C for 5 min, followed by 30 cycles of 94 °C for 1 min, 48 °C for 2 min, and 72 °C for 2 min, and a final extension at 72 °C for 5 min. Sequencing libraries were generated by the Illumina platform at Igenebook Co. (Wuhan, China). Library quality was evaluated on the Qubit 2.0 fluorometer (Thermo Scientific) and the Agilent Bioanalyzer 2100 system.

### Colony-forming units (CFU) by agar plating

Flies were briefly dipped in ethanol and dried on Kimwipe paper. Intact guts from 5 to 7 animals were dissected and homogenized in sterile phosphate-buffered saline (PBS) solution using a sterile pestle. The resulting suspension was then diluted and plated on nutrient agar (213000, BD biosciences), and incubated at 30 °C for 48 h. Colonies on each plate were counted, and the colony-forming units (CFU's) were calculated. Biological triplicates were performed for each condition. CFU values were compared statistically using Student's $t$ test, and graphs were plotted in GraphPad Prism 6.

### ROS measurement

For DHE staining, intestines were freshly dissected in Schneider's medium and incubated in 30 µM DHE (Cat# D11347, Invitrogen) for 5 min at room temperature in the dark. After two washes in 1× XPBS, the samples were mounted and immediately imaged. Images were captured immediately using a Zeiss AxoImager M2 with the apotome system (543 nm excitation, 550–610 nm detection).

### ATP assay

Fly guts were dissected and homogenized in 100 µL PBS containing 4 mM EDTA and protease inhibitor (Roche) on ice. After centrifuging at 12,000 × $g$ for 10 min at 4 °C. The supernatant was transferred into a new tube and boiled for 5 min. In total, 20 µL supernatant was diluted in 80 µL double distilled water and then mixed well with 100 µL CellTiter-Glo® Reagent (Promega, #G7573) in a 96-well plate. The plate was incubated at room temperature for

10 min, and the luminescence value was measured by a microplate reader (Synergy HTX, BioTek, Winooski, Vermont, USA). Each reading was normalized to protein concentration.

## Drosophila food intake measurement

Food intake was measured by the capillary feeder assay (CAFÉ) with modifications. Around 15–20 sex-matched flies (3–4 days old) were dry starved for 4 h before feeding with liquid food via the U-GLAD system (Liang et al, 2020). The amount of liquid food consumed by flies was measured after 1 h, food was colored with blue food dye (Erioglaucine disodium salt, MACKLIN, Cat#3844-45-9) for visualization. The volume decrease at each time point was calculated.

## Western blot analysis

For protein detection, samples were collected, and western blot analysis was conducted as previously described (Ma et al, 2021). Primary antibody ATP5A (#ab14748, Abcam) were incubated at 1:2000, and actin antibody (#A2066, Sigma) for 1:10000. Predicted molecular weight: ATP5A 53kD; Actin 42kD.

## Blood glucose, oral glucose tolerance test (OGTT), intraperitoneal insulin tolerance test (ITT) in mice

Blood glucose was measured using a tail vein prick and a glucometer (Yuwell, YH-550). OGTT was performed following the final type 2 diabetic model-building period. The mice were fasted for 14-h before the OGTT and then were treated with glucose (2 g/kg body weight) orally. Blood glucose concentration was measured at 0, 30, 60, 120 min by tail vain blood after gavage using a glucometer. After building the diabetic model mice and the 3-week 0.04% TA treatment, ITT was performed to assess the insulin resistance following 6-h fasting. Mice received 0.75 unit/kg of regular insulin (Macklin, I860440) by i.p. injection and we measured blood glucose levels at 0, 20, 40, 60, and 90 min after insulin injection.

## Food intake and body weight in mice

C57BL/6 male mice and the standard chow were supplied by the laboratory animal center, Tongji University, China. Mice were housed in the Specific Pathogen Free (SPF) facility ($21 \pm 1\,°C$, $55 \pm 5\%$ relative humidity, a 12-h light/dark cycle). Animals were allowed free access to water and the standard chow for at least 1 week prior to starting the experiments.

All experimental procedures were carried out in accordance with the internationally accepted principles for laboratory animal use and care, and approved by the Animal Ethics Committee, Tongji University, China (TJAB05320101).

C57BL/6 male mice of two groups were fed with regular chow diet (Control) or with high-fat diet (HFD) ad libitum for 12 weeks. The Control and HFD groups were without treatment or received tyramine in the drinking water during 4 weeks under the form of a 0.04% solution that was changed weekly. High-fat diet (D12492) and chow diet was purchased from Xietong Organism. The consumed chow and body weight were measured daily. Animals were weighed and euthanized after being fasted overnight at the endpoint of the treatments.

## TC (total cholesterol) and liver TAGs measurement in mice

Blood samples were collected after an overnight fast 4 weeks after treatment with TA. Serum triglyceride (A110-1-1, Nanjing Jiancheng Bioengineering Institute) and total cholesterol (A111-1-1, Nanjing Jiancheng Bioengineering Institute) concentrations were determined by enzymatic methods.

The triglyceride content of the liver was determined. Briefly, 100 mg of tissue was homogenized and extracted with 2 mL of isopropanol. After centrifugation (3000 rpm), the triglyceride content of the supernatant was determined (A110-1-1, Nanjing Jiancheng Bioengineering Institute).

## Statistical analysis

Data were plotted and analyzed using GraphPad Prism 8 (GraphPad, San Diego, CA, (USA)) and reported in figure legends. Experimental flies and genetic controls were tested in the same condition, and data are from at least three independent experiments. Student t test or two-way ANOVA analysis was used for statistics. $P$ values < 0.05 were considered significant; significance values are indicated as $*P < 0.05$; $**P < 0.01$; $***P < 0.001$; $****P < 0.0001$. All pooled data are presented as mean ± standard error of the mean (SEM). For details on the number of technical and biological replicates, please refer to the individual figure legends.

## Chromatin immunoprecipitation (ChIP)

Thirty intestines were homogenized with a pestle on ice and further sonicated using the Covaris system (Gene Co, M220) plus system. Cross-linked DNA fragments were then immunoprecipitated with the p-CREB antibody or equivalent concentrations of normal rabbit IgG as a negative control. The DNA was then eluted from the immune complexes (ChIP kit, 17-10460, Merck, Darmstadt, Germany) and subjected to PCR amplification of the region of the *Magro* gene promoter containing the CRE fragments (CRE-1 and CRE-2) using the following primer sets: CRE-1: (F): CTTTACCACCCTATTATCTTAAC, (R): TCTATTCCGCTTGC TAAGTG and CRE-2: (F): CAGTCGTAGTTTCTTCAGATG, (R): AATGGTCTCAACTGGATTC.

## RNAseq analysis

For RNAseq in fly guts, around 20 guts from each sample were dissected in RNase-free PBS and placed in Trizol. The extracted RNA and cDNA library were generated as previously described (Dutta et al, 2013). Sequencing was performed on an Illumina HiSeq2000 machine. Data were analyzed using the OmicShare software. CRE motif in the promoter region (up to 2 kb upstream of TSS) of DEGs were predicted by TRANSFAC–geneXplain (https://genexplain.com/transfac/).

## Quantification of mtDNA

To quantify mtDNA levels, the total DNA of the indicated genotype was isolated from adult fly guts using a DNeasy Blood & Tissue Kit (QIAGEN). One-step quantitative real-time PCR was performed on a CFX96 PCR system (Bio-Rad) using SYBR Green qPCR SuperMix (Applied Biological Materials) with primers against mtDNA 16 S and

RP49 as described previously. mtDNA 16 S (forward: AAAAAGA TTGCGACCTCGAT, reverse: AAACCAACCTGGCTTACACC).

## Fluorescent fatty acid tracking in enterocytes

The metabolic flux of fatty acids in enterocytes was examined by fluorescently labeled fatty acids, which have been used in adipocytes in previous studies (Kolahi et al, 2016). BODIPY-FL $C_{12}$ (BODIPY-$C_{12}$, Molecular Probes, D3822) was diluted at 10 µM. BODIPY-FL-C12 was mixed with ND and fed to flies for 1 h, The guts and abdominal fat body were then dissected and fixed in 4% formaldehyde for 30 min. After washing in washing buffer for 1 h, the samples were co-stained with LipidTOX (CY5, Invitrogen, H34477) to label neutral lipids. After 30 min, the samples were mounted for imaging on an Olympus SpinSR10 microscope.

## Data availability

All sequencing datasets generated in this study are freely available through the China National Center for Bioinformation (CNCB), with transcriptome accession number CRA014780, 16 S rDNA sequencing accession number CRA015592, metabolome accession number OMIX006080. All data generated or analyzed during this study are available from the corresponding author upon request.

The source data of this paper are collected in the following database record: biostudies:S-SCDT-10_1038-S44318-024-00162-w.

## Peer review information

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

## Acknowledgements

The authors thank Bloomington Drosophila Stock Center, Vienna Drosophila Stock Center, Tsinghua Fly Center and Dr. Hai Huang, Dr. Heinrich Jasper for stocks and reagents. We also thank Dr. Ke Wei for the helpful discussions. This work was supported by a National Key Research and Development Project [2018YFA0107100], National Natural Science Foundation of China [grant no. 31871371 and 32071147] to HD.

## Author contributions

**Peng Ma**: Conceptualization; Resources; Data curation; Formal analysis; Validation; Investigation; Methodology. **Yao Zhang**: Methodology. **Youjie Yin**: Investigation; Methodology. **Saifei Wang**: Validation; Investigation; Visualization. **Shuxin Chen**: Validation. **Xueping Liang**: Methodology. **Zhifang Li**: Resources. **Hansong Deng**: Conceptualization; Supervision; Funding acquisition; Validation; Investigation; Project administration.

Source data underlying figure panels in this paper may have individual authorship assigned. Where available, figure panel/source data authorship is listed in the following database record: biostudies:S-SCDT-10_1038-S44318-024-00162-w.

## Disclosure and competing interests statement

The authors declare no competing interests.

# Expanded View Figures

**Figure EV1.   Related to Fig. 1, cytoCa$^{2+}$ in enterocytes was activated by HFD to reduce lipid levels through the Gαq/PLCβ/IP3R cascade.**

(A) CytoCa$^{2+}$ levels in ECs of the R2 region. Subcellular localization of Ca$^{2+}$ signals (GCaMP3, green), and F-actin (LifeactRFP, red) in ECs of the R2 region were shown under confocal microscopy. Insets are high-mag images of the boxed area. Top: Sagittally view, bottom superficially view. Scale bars: 10 μm. Genotype: UAS-GCaMP3; NP1Gal4$^{ts}$; UAS-LifeactRFP. (B) Distribution of neutral lipid levels (Lipid$^{TOX}$, red) and cytoCa$^{2+}$ levels (GCaMP3, green) were examined in the R2 region. Representative images are shown. Scale bars: 50μm. Genotypes: UAS-GCaMP3; NP1Gal4, tubGal80$^{ts}$; UAS-*Serca$^{RNAi}$* or UAS-GCaMP3; NP1Gal4, tubGal80$^{ts}$; UAS-*IP3R$^{RNAi}$*. (C, D) Overexpressing Stim and Orai in ECs is sufficient to reduce lipid content in the gut and whole body. Neutral lipid levels were measured by a TAG kit. Triplicates were performed for statistical purposes. *t* test for statistics, mean ± SEM is shown. *$P$ < 0.05. Genotype: UAS-GCaMP3; NP1Gal4, tubGal80$^{ts}$; UAS-Stim, UAS-Orai. (E) Related to Fig. 1G, Intestinal neutral lipids were increased by coconut oil (30%) containing HFD. Representative images were shown. Scale bar: 100 μm. (F) Intestinal neutral lipids were also increased by lard oil (30%) containing HFD. ORO intensity in the R2 region was quantified. Three independent experiments were performed. $n > =10$ animals for each experiment. Mean ± SEM were shown, *t* test for statistics. **$P$ < 0.01. (G) cytoCa$^{2+}$ in ECs was recorded after feeding with lard oil containing HFD for 4 days. cytoCa$^{2+}$ was quantified as relative fluorescence ratio of GCaMP5 vs. tdTomato. At least 80 enterocytes from 6 guts were quantified. Each dot represents one EC. *t* Test for statistics. ****$P$ < 0.0001. Genotype: NP1Gal4, tubGal80$^{ts}$; *UAS-tdTomato-P2A-GCaMP5G*. (H, I) Knockdown of *IP3R* (H) or *Gaq* (I) in ECs increased ORO intensities in guts under both ND and HFD conditions. Three independent experiments were performed. $n = 6$ animals for each experiment. Two-way ANOVA analysis for (H), *t* test for (I). Mean ± SEM were shown., *$P$ < 0.05, **$P$ < 0.01, ***$P$ < 0.001.

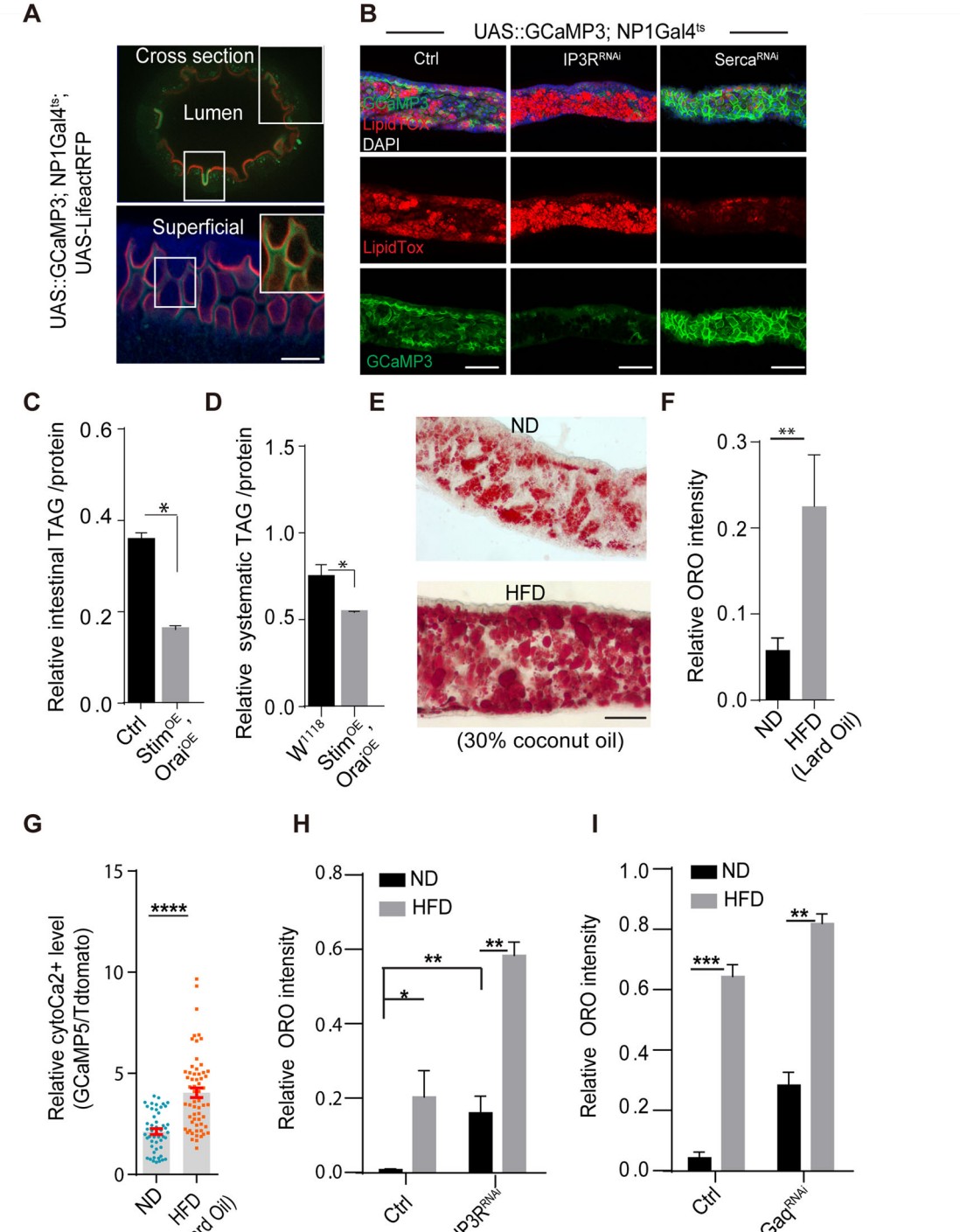

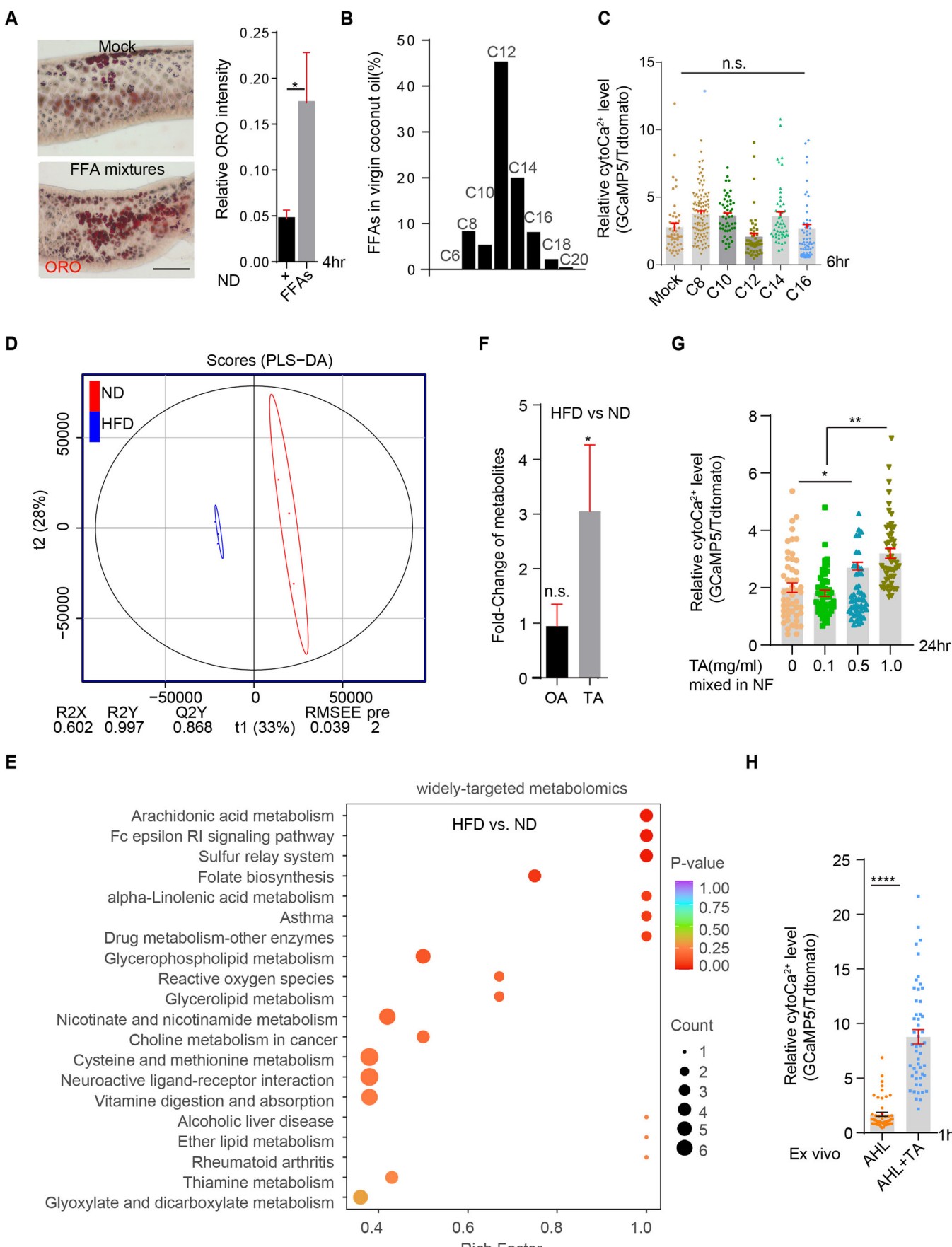

**Figure EV2.   Related to Fig. 2: Tyramine induced by HFD promotes cytoCa²⁺ levels in enterocytes.**

(**A**) Intestinal lipid levels were increased when flies were fed ND supplemented with mixtures of fatty acids derived from coconut oil for 4 h. Representative images of ORO staining are shown on the left, and quantifications on the right. Three independent experiments were performed. $n = 6$ animals for each condition. Mean ± SEM were shown. $t$ test for analysis. $*P < 0.05$ Scale bar: 200 μm. (**B**) Medium-chain fatty acid composition of virgin coconut oil. Modified from an earlier publication (Sacks, 2020). (**C**) Flies fed with ND supplemented with the indicated saturated fatty acids for 6 h failed to increase enteric cytoCa²⁺. At least 80 enterocytes from 6 guts were quantified. $t$ Test for analysis. Mean ± SEM were shown. n.s.: no significance. Genotype: NP1Gal4, tubGal80ᵗˢ; *UAS-tdTomato-P2A-GCaMP5G*. (**D**) The supervised multivariate method of partial least squares-discriminant analysis (PLS-DA) was used to maximize the metabolome differences between sample pair. PLS-DA score plots generated from PLS-DA models in HFD and ND. R2X = 0.602, R2Y = 0.997, Q2Y = 0.868; RMSEE represent Root Mean Square Error of Estimation. The scores plot of PLS-DA modeling shows a different clustering tendency. (**E**) The vertical axis represents the enriched KEGG classification. The horizontal axis is the rich factor (rich factor ≤1), which represents the ratio of the number of differentially expressed proteins to those identified in the KEGG pathway. The size of the circular area represents the number of differentially expressed proteins, and the circular color represents the enrichment $P$ value of the differentially expressed proteins under the KEGG classification. (**F**) Widely targeted metabolomics indicated that tyramine (TA) readings are significantly increased while octopamine (OA) readings are largely unchanged. $n = 3$, mean ± SEM. $t$ test for statistics, n.s.: no significance, $*P < 0.05$. (**G**) CytoCa²⁺ in ECs was examined after flies were fed diets supplemented with TA at the indicated concentrations. Two-way ANOVA analysis for statistics. mean ± SEM is shown, $*P < 0.05$, $**P < 0.01$. At least 60 enterocytes from 4 guts were quantified. Genotype: NP1Gal4, tubGal80ᵗˢ; *UAS-tdTomato-P2A-GCaMP5G*. (**H**) CytoCa²⁺ level in ECs from freshly dissected guts was increased after incubation with AHL plus TA (1 mg/ml) for 1 h. Approximately 50 ECs from 4 intestines were analyzed. $t$ test for statistics, mean ± SEM, $****P < 0.0001$.

                                                           

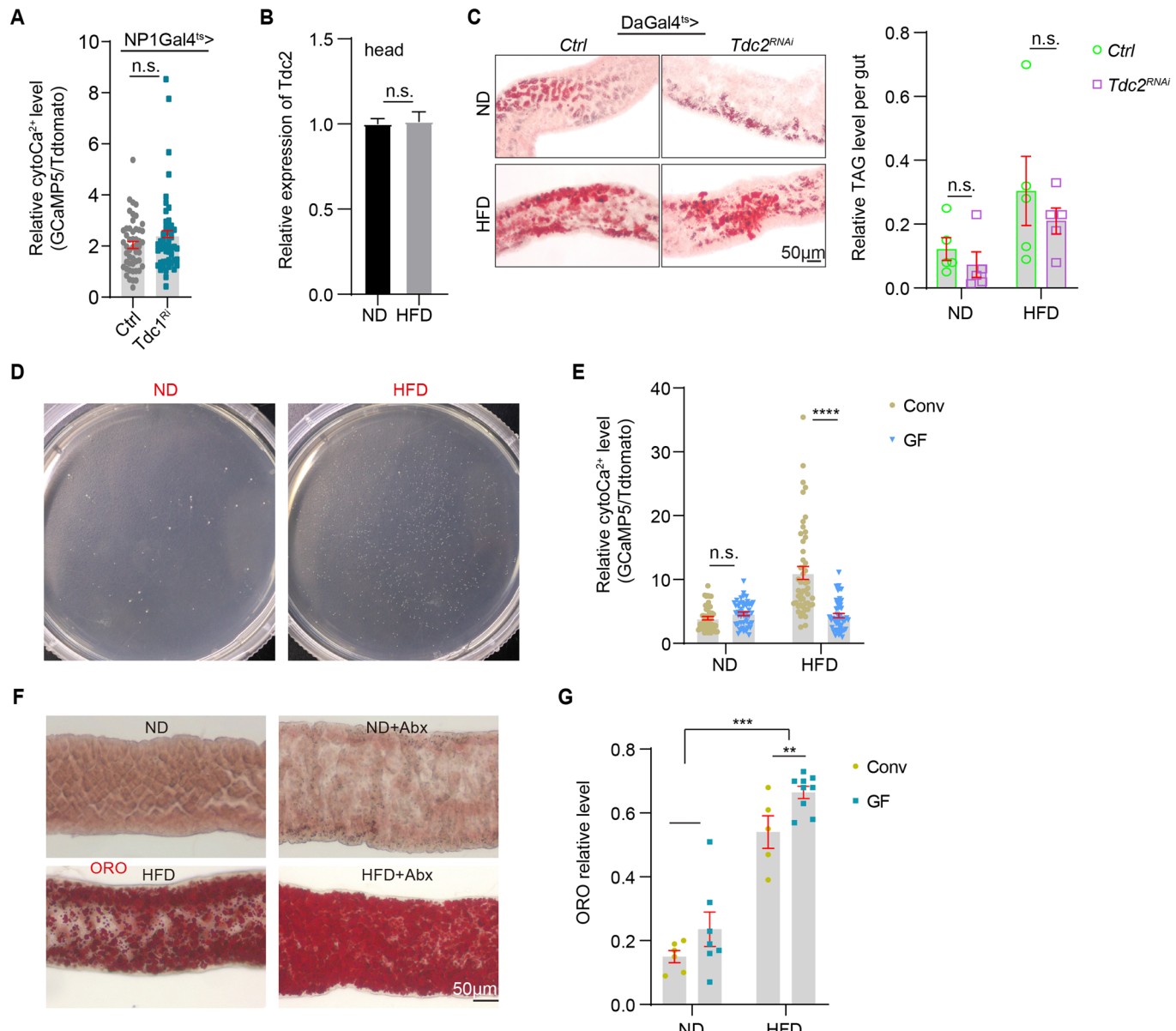

**Figure EV3. Related to Fig. 3: Bacterial load in fly guts was increased by HFD.**

(A) cytoCa2+ in ECs was examined when *tdc1* was temporally silenced in ECs by NP1Gal4. Approximately 50 ECs from 4 intestines were analyzed. *t* Test for statistics, n.s.: no significance, mean ± SEM. Genotype: NP1Gal4, tubGal80ts; *UAS-tdTomato-P2A-GCaMP5G*. (B) *Tdc2* expression in fly head was examined by RT-qPCR. Means ± SEM from biological triplicates. Student *t* Test for statistics, n.s: no significance. (C) Lipid levels (indicated by ORO staining) in ECs were examined when *tdc2* was ubiquitously inhibited. Representative images shown on left and quantifications on the right. Biological triplicates performed for statistics. $n = 5$ for each condition. *t* test for statistics, mean ± SEM shown, n.s.: no significance. Scale bar: 50 μm. Genotype: DaGal4, tubGal80ts; *UAS-tdc2RNAi*. (D) HFD increases gut bacterial load. 50 μl of homogenized sample from gut lysates of 6 animals were plated onto each antibiotic-free nutrition agar plate. (E) Ca²⁺ levels in ECs are also significantly reduced in germ-free flies under HFD conditions. Relative cytoCa²⁺ levels in ECs were monitored by two-photon live imaging. Each dot represents one EC, at least 100 ECs from 6 guts were quantified for each condition. n.s.: no significance, ***$P < 0.001$. (F) Related to Fig. 3C, antibiotic cocktail (Abx) treatment further increases neutral lipid content in the intestine. Representative images of ORO staining in the R2 region are shown. Scale bar: 50 μm. (G) Germ-free (GF) flies contains more intestinal lipid content under HFD conditions than those raised under conventional condition (Conv). Lipid levels were examined by ORO staining. Biological triplicates performed. 6–8 animals analyzed for each condition. *t* test for statistics, mean ± SEM shown. **$P < 0.01$, ***$P < 0.001$.

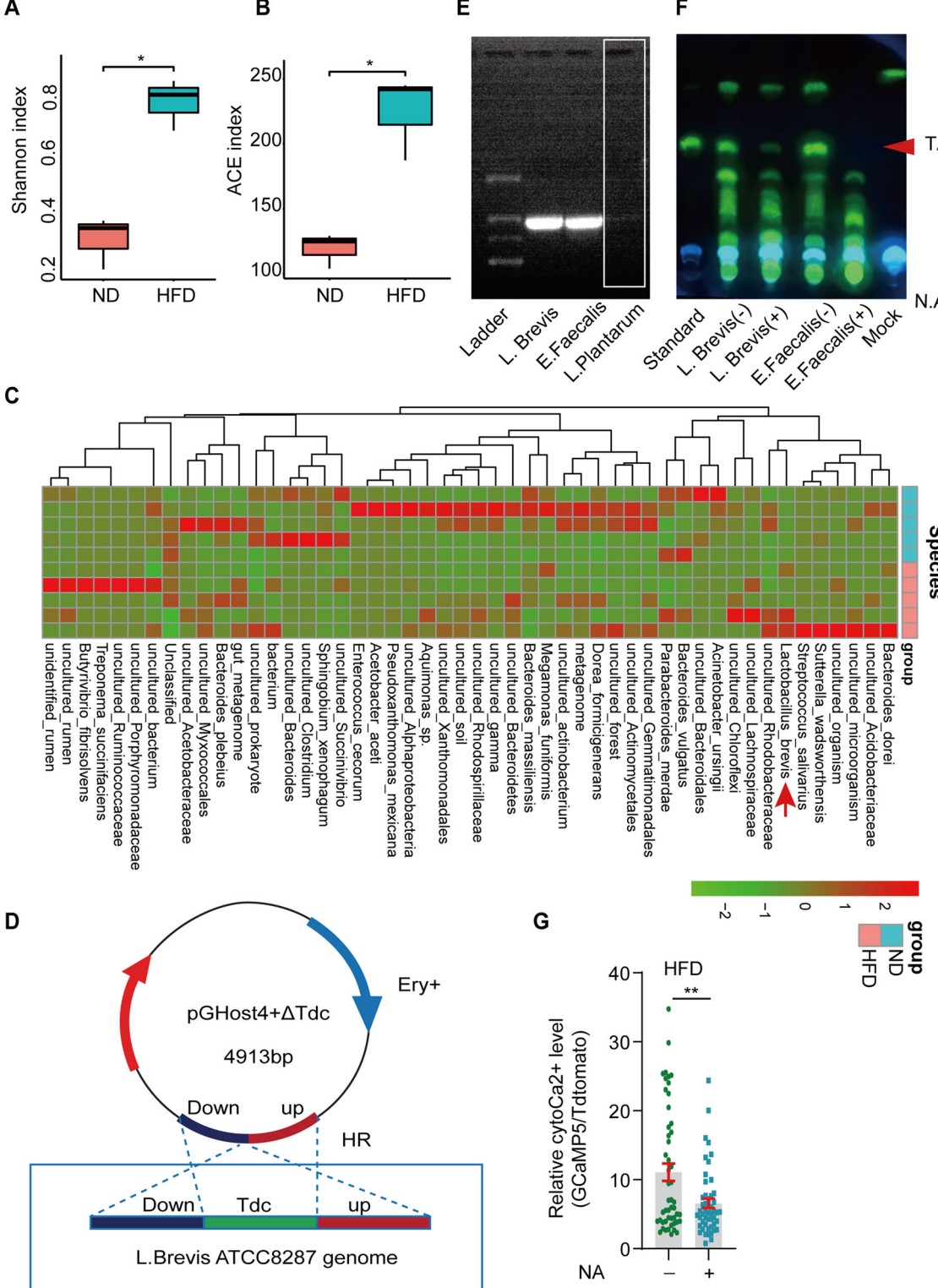

**Figure EV4.  Related to Fig. 3, Gram-positive bacteria increased by HFD is responsible for tyramine production.**

(A, B) The Shannon index and ACE index of Krushal–Wallis analysis of 16 S amplicon sequencing were used to plot species diversity and abundance respectively. *$P < 0.05$, $n = 5$. For Shannon index, The minima, maxima, center(median), and IQR(interquartile range) are marked in boxplots. (C) Heatmap of differentially expressed bacterial family in HFD v.s. ND was shown. The arrow points to the *Lactobacillaceae Brevis* (*L. Brevis*) species, which is significantly increased in the HFD condition. (D) Design for the generation of *tdc* deletion mutants in *L. Brevis* based on the homologous recombination (HR) strategy. For details, see "Methods". (E) Electrophoresis gel analysis from PCR using primer set for *tdc* gene. The product (941 bp in length) was not present in the *L. Plantarum* lane (the boxed one). (F) Thin-layer chromatography (TLC) showed that nicotine acid (N.A., 0.5 mg/ml) robustly reduced TA levels (red arrowhead) in the culture supernatant of *L. brevis* and *E. faecalis*. (G) The cytoCa$^{2+}$ level in ECs was examined when flies were fed with N.A. (0.5 mg/ml) under HFD conditions. Approximately 40 ECs from 4 intestines were analyzed. *t* Test for statistics, **$P < 0.01$, mean ± SEM shown. Genotype: NP1Gal4, tubGal80$^{ts}$; *UAS-tdTomato-P2A-GCaMP5G*.

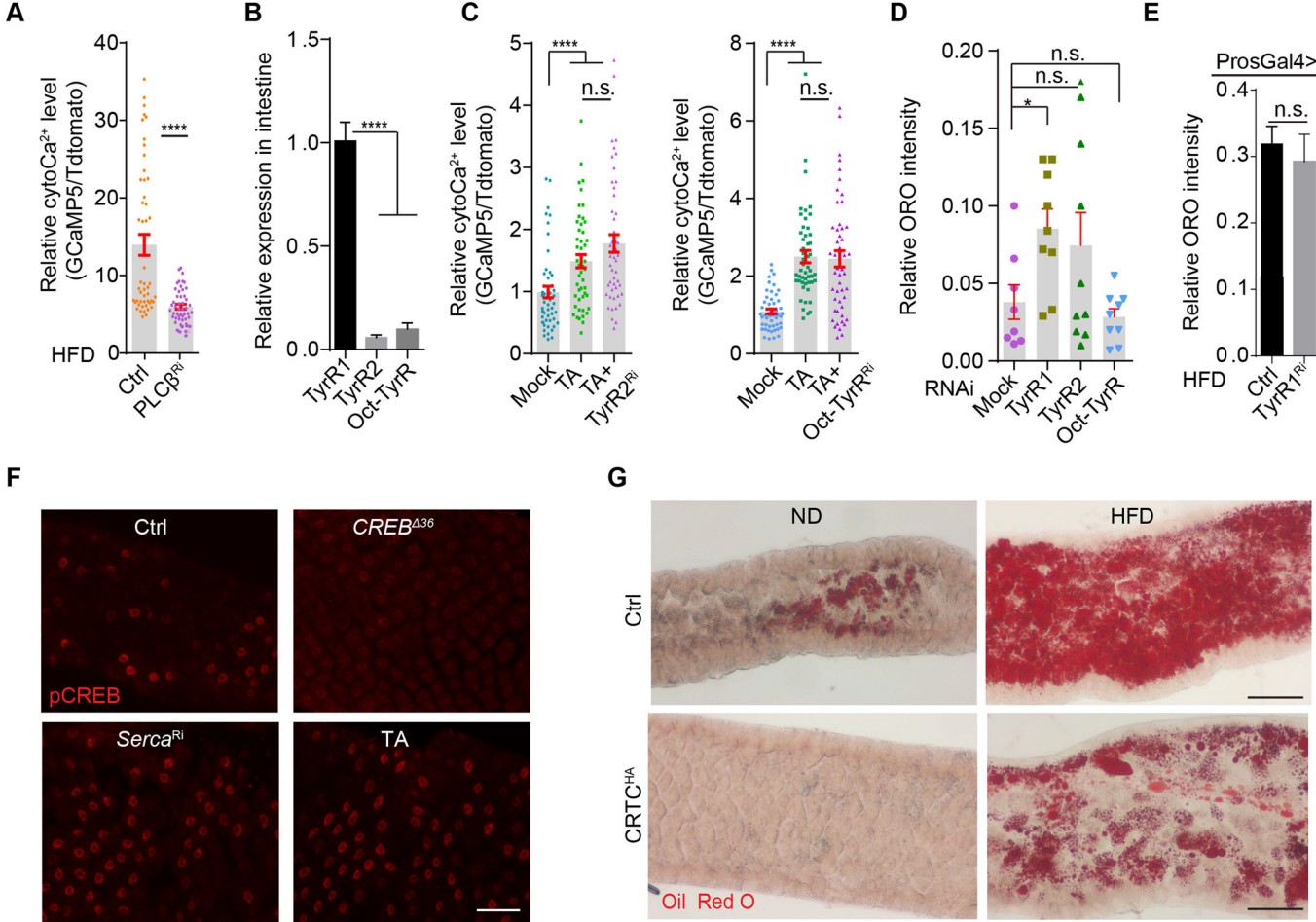

**Figure EV5. Tyramine activates enteric Ca²⁺ signaling through tyramine receptor, *TyrR1*.**

(A) PLCβ knockdown in ECs significantly reduces HFD-induced cytoCa²⁺ levels. Each dot denotes one EC, at least 60 ECs from 5 guts were quantified. Student *t* test for statistics. Mean ± SEM were shown. ****$P < 0.0001$, Genotype: NP1Gal4, tubGal80ᵗˢ; *UAS-tdTomato-P2A-GCaMP5G, UAS-PLCβ^RNAi*. (B) Relative expression of tyramine receptors in fly guts by qRT-PCR. Means ± SEM from biological triplicates. Student *t* Test for statistics, ****$P < 0.0001$. (C) Knockdown of *TyrR2* or *Oct-TyrR* in ECs failed to block TA-elevated cytoCa²⁺ levels. CytoCa²⁺ in ECs was recorded when freshly dissected guts were incubated with AHL containing TA(1 mg/ml). Biological triplicates were performed, mean ± SEM were shown. One-way ANOVA analysis for statistics, ****$P < 0.0001$. n.s. no significance. Genotypes: NP1Gal4, tubGal80ᵗˢ; *UAS-tdTomato-P2A-GCaMP5G, UAS-Oct-TyrR^RNAi* or NP1Gal4, tubGal80ᵗˢ; *UAS-tdTomato-P2A-GCaMP5G, UAS-TyrR2^RNAi*. (D) Knockdown of *TyrR1* instead of *TyrR2* or *Oct-TyrR* in ECs significantly increases intestinal neutral lipids. Three independent experiments were performed. $n = 10$ animals for each experiment. Two-way ANOVA analysis for statistics. Mean ± SEM was shown, *$P < 0.05$, n.s. no significance. ORO intensity of R2 region was quantified for each condition. Genotypes: NP1Gal4, tubGal80ᵗˢ; *UAS-TyrR1^RNAi*, NP1Gal4, tubGal80ᵗˢ; *UAS-Oct-TyrR^RNAi*, or NP1Gal4, tubGal80ᵗˢ; *UAS-TyrR2^RNAi*. (E) Intestinal neutral lipids was examined by ORO staining when *TyrR1* was silenced in enteroendocrine cells by ProsperoGal4. Genotypes: ProsGal4, tubGal80ᵗˢ; *UAS-TyrR1^RNAi*. Biological triplicates were performed. Student t test for statistics, mean ± SEM was shown, n.s. no significance. (F) Related to Fig. 4H,I, CREB activities in ECs were quantified by fluorescence intensity of anti-p-CREB. Representative images were shown. Scale bar: 20μm. Genotypes: W^1118 or CREB^Δ36, or NP1Gal4, tubGal80ᵗˢ; *UAS-Serca^RNAi*. (G) Related to Fig. 4L, CRTC overexpression suppressed HFD-induced intestinal lipid accumulation. Representative images of ORO staining in R2 region were shown. Genotypes: NP1Gal4, tubGal80ᵗˢ or NP1Gal4, tubGal80ᵗˢ; *UAS-Serca^RNAi*. Scale bar: 200 μm.

