## [Peer Review File · The EMBO Journal]

Gut microbiota metabolite tyramine ameliorates high-fat diet-induced insulin resistance via increased Ca²⁺ signaling

Peng Ma, Yao Zhang, Youjie Yin, Saifei Wang, Shuxin Chen, Xueping Liang, Zhifang Li, and Hansong Deng

Corresponding author: Hansong Deng (hdeng@tongji.edu.cn)

Review Timeline:

Submission Date:	11th Oct 23
Editorial Decision:	14th Nov 23
Revision Received:	29th Mar 24
Editorial Decision:	7th May 24
Revision Received:	7th Jun 24
Editorial Decision:	19th Jun 24
Revision Received:	20th Jun 24
Accepted:	21st Jun 24

Editor: Ieva Gailite

Transaction Report:

Dear Hansong,

Thank you for submitting your manuscript for consideration by the EMBO Journal. We have now received comments from three reviewers, which are included below for your information.

As you will see from the reports, all reviewers find the proposed role of microbiota-derived tyramine in regulation of calcium signalling and insulin resistance per se of interest. However, they also raise a number of important concerns that would need to be addressed before they can recommend acceptance of the manuscript.

Based on the interest expressed in the reports, I invite you to address these issues in a revised version of the manuscript. I think it would be helpful to discuss the revision in more detail via email or phone/videoconferencing - please let me know which option you prefer. I should also add that it is The EMBO Journal policy to allow only a single major round of revision and that it is therefore important to resolve the main concerns at this stage.

We generally allow three months as standard revision time, which can be extended to six months in the case of major revisions. As a matter of policy, competing manuscripts published during this period will not negatively impact on our assessment of the conceptual advance presented by your study. However, please contact me as soon as possible upon publication of any related work to discuss the appropriate course of action. Should you foresee a problem in meeting this deadline, please let us know in advance to discuss an extension.

When preparing your letter of response to the referees' comments, please bear in mind that this will form part of the Review Process File and will therefore be available online to the community. For more details on our Transparent Editorial Process, please visit our website: <https://www.embopress.org/page/journal/14602075/authorguide#transparentprocess>. Please also see the attached instructions for further guidelines on preparation of the revised manuscript.

Please feel free to contact me if you have any further questions regarding the revision. Thank you for the opportunity to consider your work for publication. I look forward to discussing your revision.

With best regards,

Ieva

- a point-by-point response to the referees' comments, with a detailed description of the changes made (as a word file).
- a word file of the manuscript text.
- individual production quality figure files (one file per figure)

- a complete author checklist, which you can download from our author guidelines (<https://www.embopress.org/page/journal/14602075/authorguide>).

- Expanded View files (replacing Supplementary Information)

We realize that it is difficult to revise to a specific deadline. In the interest of protecting the conceptual advance provided by the work, we recommend a revision within 3 months (12th Feb 2024). Please discuss the revision progress ahead of this time with the editor if you require more time to complete the revisions.

Referee #1:

In this manuscript, Ma et al investigate the gut microbiome and its metabolites in the suppression of obesity and insulin resistance. Surprisingly, while genetic manipulation aimed at increasing cytoplasmic calcium negatively regulated lipid droplet accumulation, cytoplasmic calcium levels were observed to increase upon high-fat diet exposure. The authors elucidated that the surge in cytoplasmic calcium resulted from the presence of tyramine, a metabolite produced by Gram-positive bacteria containing tyrosine-decarboxylase. This metabolite was found to activate calcium signaling in enterocytes through the engagement of the TyrR1 pathway. Subsequently, the activated calcium signaling cascade induced the transcription of a lipase and genes related to mitochondrial biogenesis, effectively suppressing lipid accumulation. Moreover, the authors discovered that both tyramine and the downstream calcium signaling pathway effectively mitigate insulin resistance in both flies and mice. Although the manuscript contains intriguing ideas and data, there are several areas that require improvement, as outlined below:

1. In Fig. 3, the authors employed conventionally reared flies with administered antibiotics to manipulate the gut microbiomes. However, the gold standard approach in the field involves the use of germ-free animals. It would be informative to investigate whether the effect of a HFD on the increase in calcium levels is absent in germ-free flies. Furthermore, assessing the impact of mono-association of *L. brevis* on calcium levels would provide valuable insights. Additionally, studying the behavior of flies mono-associated with an *L. brevis* mutant that deletes the *Tdc* gene could offer further clarity, especially in comparison to germ-free flies.
2. In line 304, the authors assert that the increase in ROS implies the activation of beta-oxidation. However, the manuscript lacks further evidence to substantiate this claim. Given that ROS production can be triggered by various factors, including inflammation, it is imperative for the authors to provide direct evidence for the activation of beta-oxidation.
3. In Fig. 7, the authors evaluated insulin resistance by demonstrating AKT-Spark puncta in enterocytes. Considering the systemic effect of the calcium signal in enterocytes, as depicted in Fig. 1f, it is crucial to assess the insulin response not only in enterocytes but also in the fat body.
4. In Fig. 7c, the authors exclusively illustrate the impact of TA and CRTG in the cases of ND and HFD. However, considering the focus on insulin resistance, it is essential to include testing the response to insulin, as depicted in Fig. 7a and b.
5. Another critical concern pertains to the speculative nature of the connection between calcium signaling and insulin signaling. To establish a more concrete link, the authors should investigate whether the recovery from insulin resistance by TA occurs through an increase in systemic or local insulin secretion, or through other alterations in the secretion of factors such as ImpL. Additionally, in the insulin resistance experiments, the authors employed the NP1-Gal4 driver, which is specific to enterocytes. However, in the discussion section, the authors reference secretory factors like GLP-1 in mice. Considering that the secretion of such factors occurs in enteroendocrine cells in the fly gut, rather than in enterocytes, it would be valuable to explore whether enteroendocrine cells have TA-dependent calcium signaling mechanisms that contribute to the improvement of insulin resistance. Investigating these aspects would provide a more comprehensive understanding of the intricate relationship between calcium signaling and insulin resistance, thereby enhancing the significance of the findings presented in the manuscript.

Referee #2:

The manuscript by Ma et al., builds on previous studies in *Drosophila* that demonstrated changes in triglycerides in intracellular Ca²⁺ signaling mutants. Whereas the earlier studies looked at this phenomenon in the fat body cells of *Drosophila*, Ma et al have investigated if there are diet induced changes in the gut enterocytes. They find that a high fat diet (HFD) leads to a steady increase in the basal cytosolic Ca²⁺ of enterocytes. Raising or reducing the basal cytoCa²⁺ by genetic means in the enterocytes correlated with changes in accumulation of lipids in the gut. They have then gone onto identify the factor from HFD that leads to changes in cytoCa²⁺ through a targeted metabolomic profiling of HFD guts and intriguingly find Tyramine. Their data then identify the source of the Tyramine as gram positive bacteria that appear to populate the gut better when flies are fed with HFD. Tyramine in turn acts through Tyramine receptors present on the gut to activate IP₃-Ca²⁺ signaling and CREB, leading to changes in gene expression. Among the genes downregulated is a dietary lipase called Magro, loss of which suppresses accumulation of intestinal lipids and results in greater excretion of TAGs. The RNAseq data also leads them to suggest that a HFD results in a switch from lipogenesis to B-oxidations and promotes mitochondrial biogenesis in the enterocytes. As a consequence of the HFD they find that reporters of Akt-activity and PI3K activity are reduced in the enterocytes and these markers can be reversed by TA feeding. Finally, the significance of these findings were tested in mice on HFD where the data are somewhat ambiguous. The discovery that gut microbiota produce Tyramine that affects intracellular Ca²⁺ signaling and Ca²⁺-mediated gene expression in the gut epithelium is exciting and well presented. Overall, there is insufficient discussion on how HFD negatively regulates diet induced obesity but yet results in diet induced obesity and loss of insulin sensitivity. I have some queries regarding the conclusions reached and a few suggested experiments that would help clarify the results.

Major points:

- 1) Abdominal fat body cells play important roles in glucose metabolism in flies. There is no data investigating the status of lipid storage in the fat body cells upon Tyramine feeding. Does TA act locally on the gut epithelial cells or does it also change fat accumulation in the fat body? Does the switch to mitochondrial biogenesis occur in the fat body as well? This should be tested.
- 2) Does the fly encoded Tdc play a role in generating TA under the HFD conditions? It should be possible to test this using an RNAi line or Tdc mutant.
- 3) From Fig 5a it is not clear if the DEGs shown classify into specific groups. Better and more detailed analysis of the RNAseq is needed including gene ontology terms. What is the measure of differential expression - this should be mentioned in the figure or its legend. How many of the DEGs contain CREB binding sites?
- 4) Magro downregulation upon higher Ca²⁺ signaling leads to reduced accumulation of intestinal lipids on HFD. More TAGs are excreted. How does HFD get converted to TAGs in the absence of Magro? Is Magro the only lipase gene with changed expression? Is there another lipase active? This needs to be explained and tested.
- 5) The comparative time scales of HFD induced high cytoCa²⁺, accumulation of lipids and loss of insulin sensitivity in enterocytes is unclear. If HFD supports tyramine producing gut bacteria then why do these not protect against HFD-induced insulin sensitivity? Why does insulin resistance develop when TA synthesizing bacteria are present in the gut? What is the role of exogenously fed TA in the HFD organisms? The authors need to address this issue.
- 6) If there is indeed a switch to mitochondrial biogenesis, levels of ATP5A2 should go up in HFD. The authors should measure this. What do mitochondria look like in HFD fed enterocytes? Is there a concomitant change in mitoCa²⁺ in HFD fed conditions?

Minor points:

- 1) Both Octopamine and Tyramine are mentioned in the text but only TA is shown in figure 2D. Both should be shown in the figure. Further the data in this figure is shown as ND vs HFD. It would be simpler if it was shown as HFD vs HD. The rationale for looking at TA is not very clear and should be explained better in the text.
- 2) CytoCa²⁺ is reduced by both GqRNAi and TyrR RNAi but the difference with TyrR RNAi is much less. Can they explain this difference? This suggests the possible involvement of yet another Gq-coupled receptor. Based on their metabolomic data can they comment on this?
- 3) Some changes would improve Fig 4. In Figs 4e-m it would be helpful if controls are consistently shown as black bars. Fig 4f should show significance between ND and HFD. Throughout the figure it is not clear which genotypes are being compared to show significance. The significance bars need to be placed more carefully and explained clearly in the legend.
- 3) In Figure 6D the BodipyC12 signals are not visible. Please show each channel separately for better clarity. The status of BODIPY-C12 in HFD should be described in the results.

4) The authors should distinguish between TAGs and fat all through the result section - the two are not synonymous.

Referee #3:

In this interesting paper, Deng and colleagues use *Drosophila* to elucidate a novel pathway in which high fat diets (HFD), via a metabolite produced by the microbiota (Tyramine; TA), alter the digestive process to make it more efficient and less stress-inducing on the gut epithelium. Their experiments test and confirm many components of this signal transduction pathway to elaborate a long string of functional interactions leading from HFD to changes in the microbiota to TA production, to the TA receptor (TyrR1) to Calcium flux to CaMKII to CREB/CRTC and finally to their transcriptional targets that alter lipid digestion, lipogenesis, and mitochondria biogenesis in gut enterocytes (summarized in Fig 7e). At the end of the paper, a bit of data is offered to make a connection to insulin resistance in both flies (Fig 7) and mice (Fig 8), though the detailed pathway analysis is confined to flies. The main point of significance and interest in my opinion lies in the role of diet-induced microbiota changes and Tyramine, which is proposed as a potential dietary supplement to combat the negative effects of HFD, including metabolic syndrome, and insulin resistance in diabetes. This is a logical proposition. The study is expertly constructed and uses a wide variety of appropriate genetic tools, sensors, drugs, and assays to test this putative pathway. It is fairly well written, and logically presented, making a rather complex study easy enough to follow. There are some typos and grammatical issues, and the Discussion has a lack of focus that suggests dependence on AI editing. I liked the story overall, and all the data are without exception consistent with the author's assertions. However, there are several links in the proposed pathway that rely either on weak (marginally significant) data, or on sets of tests that are not entirely rigorous. For these reasons (detailed below), my confidence in the overall pathway being accurate and true is not especially high. However I believe the authors should be able to bolster the weak links in their pathway with additional data that would make the study appropriate for publication.

Major concerns:

1. The weakest apparent link in the author's proposed pathway is the role of microbiota changes, and microbiota-produced Tyramine (TA) as a regulator of gut enterocyte functions. The data relevant to this (Fig 2, 3, S4) is mostly associative, and more rigorous tests should be performed to confirm that it is actual TA from specific bacteria that is the critical secreted signaling in this system. Given the data presented (notwithstanding the requirement of TyrR1) it is possible that another bacterial metabolite executes the function in question. The authors rely heavily on TA supplementation at 1mg/ml in many experiments, and it is not determined whether this (extremely high) amount of TA has any physiological relevance. What is the concentration of TA produced by *L. Brevis* or *L. Fecalis* in the gut lumen? One standard, elegant type of experiment that could help cement this part of the story is to knock out TA production genes in the bacteria, and then show that the bacteria fail effect downstream gut functions, which can be rescued by TA supplementation (at physiological levels similar to those produced by gut bacteria.)
2. The experiments with bacteria in combination with HFD are limited to Fig 3 and don't test the effects of the microbiota on downstream effects described in Figs 4-7 (e.g. on CREB & CREB targets), or in mice. Adding such experiments could increase confidence in the authors' overall conclusions.
3. The data in Fig 3d don't strongly associate HFD with TA-producing microbiota. Additionally, the heat maps (Fig 3d, S4) doesn't specific species, so it is impossible to tell if *L. Brevis* and *L. Fecalis*, used in later experiments, are increased by HFD.
4. The data in Fig 3a show a massive increase in total gut bacteria amount (CFU) from HFD. This raises the question of whether it is just more bacteria that cause the effect, and not different bacteria. Can the authors rule this out? Overall, the section about bacteria should be improved.
5. Many key experiments have rather high p values, in the range .05-.01 (one *). I counted 9 examples in Fig 1-7. Since low confidence results are additive in a linear pathway, this brings the probability of the whole pathway down considerably. For instance if nine results with a probability of being correct of 95% must all be correct for the pathway to be correct, then confidence in the whole pathway being correct is $.95^9=.63$. That is not very good. Hence, the authors should strive to get more data points, or more accurate data in order to lower their p values for all key experiments to $>.01$ (**). This is also true for the mouse data in Fig 8b,d.
6. The effect on magro gene expression is shown only for mRNA (Fig 5). Since mRNA levels do not always correlate with protein activity levels, data should be provided showing either the protein expression of this lipase, or levels of its enzymatic activity.
7. The important lipid metabolism experiment in Fig 6d is apparently done only in the +/- CRTC condition, and not with HFD or TA administration or microbiota changes. These results should be extended and quantified. Otherwise the metabolic endpoint effects of the pathway are not so clear. Also, the sentence explaining this experiment is non-sensical and needs to be corrected (line 311-313.)
7. The mouse data (Fig 8) are very superficial, and don't actually demonstrate dysbiosis from HFD in mice.

Minor concerns:

1. Raw metabolomics and transcriptomics data needs to be made available to readers.
2. On line 158, please note that the short time interval (4h) is meant to help show immediate effects in proximal signaling.
3. Results from experiments should be presented in the past tense, not the present tense. Results are logically pertinent and true only to the experiment and experimental conditions at the time the experiment was performed, in the past.
4. Not all data is presented as dot plots, but it should be. Simple bar graphs with error bars are not acceptable, since these obscure n values and data scatter.
5. Fig 4a, b are missing the control conditions, with Gaq-Ri and TyrRa-Ri alone (without TA addition). These should be included.
6. Line 313 mentions "data not shown", which needs to be shown.

Referee #1:

In this manuscript, Ma et al investigate the gut microbiome and its metabolites in the suppression of obesity and insulin resistance. Surprisingly, while genetic manipulation aimed at increasing cytoplasmic calcium negatively regulated lipid droplet accumulation, cytoplasmic calcium levels were observed to increase upon high-fat diet exposure. The authors elucidated that the surge in cytoplasmic calcium resulted from the presence of tyramine, a metabolite produced by Gram-positive bacteria containing tyrosine-decarboxylase. This metabolite was found to activate calcium signaling in enterocytes through the engagement of the TyrR1 pathway. Subsequently, the activated calcium signaling cascade induced the transcription of a lipase and genes related to mitochondrial biogenesis, effectively suppressing lipid accumulation. Moreover, the authors discovered that both tyramine and the downstream calcium signaling pathway effectively mitigate insulin resistance in both flies and mice. Although the manuscript contains intriguing ideas and data, there are several areas that require improvement, as outlined below:

We thank the reviewer for the positive comments and valuable suggestions. We have addressed the reviewer's concerns point by point as follows.

1. In Fig. 3, the authors employed conventionally reared flies with administered antibiotics to manipulate the gut microbiomes. However, the gold standard approach in the field involves the use of germ-free animals. It would be informative to investigate whether the effect of a HFD on the increase in calcium levels is absent in germ-free flies.

In this study, antibiotics were used to temporarily eliminate the gut microbiome in HFD-induced obese flies. Germ-free flies required embryo bleaching and eliminating microbes from the embryonic stage on, it would not be able to address microbiota shift in obese flies. Meanwhile, the *Drosophila* microbiome has been shown to modulate developmental and metabolic homeostasis in the host (PMID:22053049, PMID:32196485), and since we're studying lipid metabolism, we didn't choose GF in the first place.

However, we agree with the reviewer that GF flies would be beneficial to further elucidate the role of bacterial contribution on Ca²⁺-mediated lipid regulation in ECs.

Consistent with Fig.3B, our new results found that Ca²⁺ levels in ECs were reduced while lipid levels in ECs were significantly increased in GF condition (**new Fig. EV3D, F**).

Furthermore, assessing the impact of mono-association of *L. brevis* on calcium levels would provide valuable insights. Additionally, studying the behavior of flies mono-associated with an *L. brevis* mutant that deletes the *Tdc* gene could offer further clarity, especially in comparison to germ-free flies.

Thanks for the suggestion. We have successfully generated a deletion allele of *Tdc* in *L. brevis* using an HR-based strategy (**new Fig. EV4D**). Germ-free flies fed with *Tdc*^{-/-} bacteria failed to increase Ca²⁺ in ECs compared to those fed with WT bacteria (**new Fig. 3E**), further indicating that TA-producing bacteria promote Ca²⁺ increase in ECs on HFD.

2. In line 304, the authors assert that the increase in ROS implies the activation of beta-oxidation. However, the manuscript lacks further evidence to substantiate this claim.

Given that ROS production can be triggered by various factors, including inflammation, it is imperative for the authors to provide direct evidence for the activation of beta-oxidation.

As the reviewer pointed out, although ROS are mainly generated in mitochondria,

infection or other stimuli can also induce ROS. Direct measurement of beta-oxidation rate using assays such as isotope labelling or Seahorse is technically challenging in *Drosophila* due to limited materials. ROS, ATP and expression and activity of key enzymes are commonly used to reflect beta-oxidation activity in flies (PMID: 34290404, PMID: 19254568). In our study, we found reduced expression of DGATs, increased fatty acid consumption and ROS production in ECs, all of which indicated that beta-oxidation was enhanced in ECs (**Fig.6**). To strengthen our conclusion, our new results also showed that ATP levels were also increased by CRTTC overexpression (**new Fig.EV7D**).

3. In Fig. 7, the authors evaluated insulin resistance by demonstrating AKT-Spark puncta in enterocytes. Considering the systemic effect of the calcium signal in enterocytes, as depicted in Fig. 1f, it is crucial to assess the insulin response not only in enterocytes but also in the fat body.

AKT-ASPARK is driven by the UAS-Gal4 system, which requires a fat body specific driver, so we used t-GPH as an indicator of insulin resistance instead. However, we didn't observe any significant changes in t-GPH signals in the fat body in HFD compared to ND (**new Fig.EV8C**).

These results suggest that the sensitivity of insulin signaling to HFD varies in different tissues. In our context, insulin signaling in gut tissue is more vulnerable to HFD than in fat body.

4. In Fig. 7c, the authors exclusively illustrate the impact of TA and CRTTC in the cases of ND and HFD. However, considering the focus on insulin resistance, it is essential to include testing the response to insulin, as depicted in Fig. 7a and b.

First, we validated that AKT-SPARK can be used as a new tool to monitor insulin sensitivity in flies (**Fig. 7a-b**). In **Fig.7c**, which is now presented in dot-plots in the revised version, we found that TA or gut-specific CRTTC overexpression significantly improved insulin resistance in ECs.

5. Another critical concern pertains to the speculative nature of the connection between calcium signaling and insulin signaling. To establish a more concrete link, the authors should investigate whether the recovery from insulin resistance by TA occurs through an

increase in systemic or local insulin secretion, or through other alterations in the secretion of factors such as ImpL.

The main goal of our study is to demonstrate how gut microbiota-derived metabolite regulates host lipid metabolism on HFD. Insulin resistance is closely associated with HFD and we used as a readout for HFD-induced metabolic syndromes. Our results shown in **Fig. 7- 8** indicated that TA or CRTC is sufficient to improve HFD-induced insulin resistance both in flies and mice.

Although insulin resistance was common in HFD-induced obesity, the detailed mechanism is not fully characterized. Insulin production, secretion, stability, receptor sensitization and alteration of the signaling cascade may all be modulated by HFD as well as by Ca²⁺ signaling. We believe that it's beyond the scope to further explore the mechanism of HFD-induced insulin resistance in this manuscript. However, we fully agree with the reviewer that this is a crucial issue for the field to tackle in the future.

Additionally, in the insulin resistance experiments, the authors employed the NP1-Gal4 driver, which is specific to enterocytes. However, in the discussion section, the authors reference secretory factors like GLP-1 in mice. Considering that the secretion of such factors occurs in enteroendocrine cells in the fly gut, rather than in enterocytes, it would be valuable to explore whether enteroendocrine cells have TA-dependent calcium signaling mechanisms that contribute to the improvement of insulin resistance. Investigating these aspects would provide a more comprehensive understanding of the intricate relationship between calcium signaling and insulin resistance, thereby enhancing the significance of the findings presented in the manuscript.

ECs make up almost 90% of the cells in the fly gut and are responsible for lipid digestion and consumption. Our results indicated that TA activates Ca²⁺ signaling in ECs, which reprogram lipid metabolism and ultimately suppress obesity and insulin resistance.

To address the reviewer's concern, we specifically knocked down *TyrR1* in enteroendocrine cells using ProsperoGal4 and found that TAG levels were largely unchanged on HFD (**new Fig.EV5E**), suggesting that TA signaling in enteroendocrine cells is not a major player in gut lipid metabolism. However, we agree with the reviewer

that it would be interesting to test in the future whether enteroendocrine signaling forms a gut-brain axis to regulate insulin signaling.

Referee #2:

The manuscript by Ma et al., builds on previous studies in *Drosophila* that demonstrated changes in triglycerides in intracellular Ca^{2+} signaling mutants. Whereas the earlier studies looked at this phenomenon in the fat body cells of *Drosophila*, Ma et al have investigated if there are diet induced changes in the gut enterocytes. They find that a high fat diet (HFD) leads to a steady increase in the basal cytosolic Ca^{2+} of enterocytes. Raising or reducing the basal cyto Ca^{2+} by genetic means in the enterocytes correlated with changes in accumulation of lipids in the gut. They have then gone onto identify the factor from HFD that leads to changes in cyto Ca^{2+} through a targeted metabolomic profiling of HFD guts and intriguingly find Tyramine. Their data then identify the source of the Tyramine as gram positive bacteria that appear to populate the gut better when flies are fed with HFD. Tyramine in turn acts through Tyramine receptors present on the gut to activate IP_3 - Ca^{2+} signaling and CREB, leading to changes in gene expression. Among the genes downregulated is a dietary lipase called Magro, loss of which suppresses accumulation of intestinal lipids and results in greater excretion of TAGs. The RNAseq data also leads them to suggest that a HFD results in a switch from lipogenesis to B-oxidations and promotes mitochondrial biogenesis in the enterocytes. As a consequence of the HFD they find that reporters of Akt-activity and PI3K activity are reduced in the enterocytes and these markers can be reversed by TA feeding. Finally, the

significance of these findings were tested in mice on HFD where the data are somewhat ambiguous. The discovery that gut microbiota produce Tyramine that affects intracellular Ca²⁺ signaling and Ca²⁺-mediated gene expression in the gut epithelium is exciting and well presented. Overall, there is insufficient discussion on how HFD negatively regulates diet induced obesity but yet results in diet induced obesity and loss of insulin sensitivity. I have some queries regarding the conclusions reached and a few suggested experiments that would help clarify the results.

We appreciate the reviewer's positive feedback. In this study, we found that the gut microbiota was remodeled as a strategy to counteract HFD-induced metabolic syndrome through bacterial-derived TA signaling. We apologize that this is not well elaborated in the Discussion section, which has now been updated in **line 418-420**.

Major points:

1) Abdominal fat body cells play important roles in glucose metabolism in flies. There is no data investigating the status of lipid storage in the fat body cells upon Tyramine feeding.

Does TA act locally on the gut epithelial cells or does it also change fat accumulation in the fat body? Does the switch to mitochondrial biogenesis occur in the fat body as well?

This should be tested.

Thanks for the suggestion. We have now tested the role of TA in the fat body under ND or HFD conditions. As shown in **new Fig.EV8D**, mitochondrial biogenesis in fat body was largely unchanged by TA under both ND and HFD conditions. Moreover, we found gut-specific knockdown of TA signaling is sufficient to suppress TAG level in fat body (**new Fig.6I**). Combined with results shown in **new Fig.EV8C**, these results indicated that TA activates Ca²⁺ signaling in ECs, which reprograms lipid metabolism and then systematically influences HFD-associated insulin resistance.

2) Does the fly encoded Tdc play a role in generating TA under the HFD conditions? It

should be possible to test this using an RNAi line or Tdc mutant.

We agree with the reviewer that this is an important issue that needs to be addressed. The *Drosophila* genome encodes two *Tdc* genes: *Tdc2* is expressed exclusively in neurons, and *Tdc1*, which is expressed non-neuronally (PMID: 15691831), was not readily detected in the adult fly gut (PMID: 23643535). Our new results showed that Ca²⁺ signaling remains largely unchanged by gut specific knocking down *Tdc1* (new **Fig.EV3A**). Moreover, expression of *Tdc2* in fly head is comparable with ND and HFD (new **Fig.EV3B**). On the other hand, we found that the TA-mediated lipid lowering effect was abolished when flies were treated with antibiotics (**Fig.7D**). Therefore, we believe that TA is mainly produced by the microbiome rather than the host under HFD conditions. The main text has been updated accordingly (**line 412-415**)

3) From Fig 5a it is not clear if the DEGs shown classify into specific groups. Better and more detailed analysis of the RNAseq is needed including gene ontology terms. What is the measure of differential expression - this should be mentioned in the figure or its legend.

How many of the DEGs contain CREB binding sites?

We have now re-analyzed the RNAseq data and present the KEGG analysis in new pane **Fig.EVS6B**. The RNAseq results have been uploaded to the open source (<https://ngdc.cnbc.ac.cn/gsa/>). Around 73% DEGs contain CRE motif as predicted by TRANSFAC (<https://genexplain.com/transfac/>). The analysis method now has been indicated in the figure legends.

4) Magro downregulation upon higher Ca²⁺ signaling leads to reduced accumulation of intestinal lipids on HFD. More TAGs are excreted. How does HFD get converted to TAGs in the absence of Magro? Is Magro the only lipase gene with changed expression? Is there another lipase active? This needs to be explained and tested.

Although the *Drosophila* genome encodes several putative lipases in midgut that may contribute to lipid digestion (PMID: 30523167), *Magro* is the only well-characterized lipase

in the *Drosophila* gut (PMID: 33716135, PMID: 34308280). Our results showed that Ca²⁺ signaling suppresses *magro* expression and lipase function, as more TAGs are excreted in fecal samples. Since *magro* transcription is not completely suppressed by CRTC, HFD can still convert into TAGs, but in a less efficient way.

We think that, like TA, silencing *magro* is a strategy for the host to counteract the syndromes associated with HFD. We have amended the main text accordingly to make it clear and precise (**line 294-295 and line 309**).

5) The comparative time scales of HFD induced high cytoCa²⁺, accumulation of lipids and loss of insulin sensitivity in enterocytes is unclear. If HFD supports tyramine producing gut bacteria then why do these not protect against HFD-induced insulin sensitivity? Why does insulin resistance develop when TA synthesizing bacteria are present in the gut? What is the role of exogenously fed TA in the HFD organisms? The authors need to address this issue.

Our BODIPY-C12 pulse-chasing experiment in **Fig.6E** showed that lipid uptake, accumulation and transport occurred very rapidly in the intestine (less than 6 hours). However, Ca²⁺ levels are not altered by incubation with fatty acids (**Fig.EV2C**) but can be modulated by the gut microbiome (**Fig.3B and new Fig EV3D**). These results support our model (**Fig.7E**) that chronic HFD feeding causes excessive intake leading to insulin resistance and dysbiosis mediated by TA-Ca²⁺ signaling. TA-Ca²⁺ signaling is beneficial for the host to counteract insulin resistance, but is not sufficient. Exogenous TA feeding would enhance this effect. We apologize again that we didn't make this clear in the previous version and have now consolidated it in the discussion section in **line 418-420**.

6) If there is indeed a switch to mitochondrial biogenesis, levels of ATP5A2 should go up in HFD. The authors should measure this. What do mitochondria look like in HFD fed enterocytes? Is there a concomitant change in mitoCa²⁺ in HFD fed conditions?

In Fig.6A-B, we found that mtDNA levels and TFAM expression in ECs were increased with CRTC overexpression. As suggested by the reviewer, we indeed found that ATP5A expression in the intestine was significantly increased in CRTC OE intestines as shown by

WB (new **Fig.6C** and new**Fig.EV7A**); Confocal images also showed more mitoGFP signals in the CRTC overexpressing ECs (new **Fig.EV7B**); Furthermore, more mitochondria were observed in HFD ECs by TEM (new**Fig.EV7C**). Taken together, these results further indicated that mitochondrial biogenesis is increased by CRTC^{OE} or HFD.

MitoCa²⁺ is mainly transferred from the ER via MAMs (mitochondrial-associated ER membrane). Since the Gq/cytoCa²⁺ pathway was activated in HFD, Ca²⁺ released from IP3R in the ER would also increase mitoCa²⁺ via MCU. We didn't assess mitoCa²⁺ in this study because mitoCa²⁺ in mitochondria plays pleiotropic roles: in addition to metabolism, mitoCa²⁺ has been shown to be involved in apoptosis, mitochondrial dynamics, mitophagy and so on. Further exploration of this part we think is stretchy for this manuscript. However, we agree with the reviewer that how subcellular Ca²⁺ crosstalk changes in response to HFD is an intriguing aspect to explore in the future.

Minor points:

1) Both Octopamine and Tyramine are mentioned in the text but only TA is shown in figure 2D. Both should be shown in the figure. Further the data in this figure is shown as ND vs HFD. It would be simpler if it was shown as HFD vs HD. The rationale for looking at TA is not very clear and should be explained better in the text.

We have now included both and rearranged the figure as suggested. We chose TA because it was consistently and significantly increased by HFD, and it's an important bioamine known to activate Ca²⁺ through its GPCR.

2) CytoCa²⁺ is reduced by both GqRNAi and TyrR RNAi but the difference with TyrR RNAi is much less. Can they explain this difference? This suggests the possible involvement of yet another Gq-coupled receptor. Based on their metabolomic data can they comment on this?

Indeed, we also noticed that the effect of TyR^{RNAi} is weaker than that of GaQ^{RNAi} in terms of Ca^{2+} signaling.

One possible reason is that the knockdown efficiency of *TyrR1* is not as strong as that of *Gaq^{RNAi}*. Alternatively, since there are three GPCRs for TA, and *TyrR1* is the main receptor for TA-mediated signaling in ECs, the other two GPCRs may also compensate by maintaining cyto Ca^{2+} when *TyrR1* is knocked down. But perhaps the most intriguing possibility is, as the reviewer mentioned, that other metabolites in the HFD condition may also activate the Gaq/GPCR cascade in ECs. We agree that this is an interesting point and have added a sentence in **line 414-417**

3) Some changes would improve Fig 4. In Figs 4e-m it would be helpful if controls are consistently shown as black bars. Fig 4f should show significance between ND and HFD. Throughout the figure it is not clear which genotypes are being compared to show significance. The significance bars need to be placed more carefully and explained clearly in the legend.

We apologize for the confusion. We have now changed them accordingly.

3) In Figure 6D the BodipyC12 signals are not visible. Please show each channel separately for better clarity. The status of BODIPY-C12 in HFD should be described in the results.

We have separated the channels as requested. And new quantification results on BODIPY-C12 on TA feeding have been included in **new Fig.6G-H and new Fig.EV7G-H**.

4) The authors should distinguish between TAGs and fat all through the result section - the two are not synonymous.

Thanks for pointing that out. Obviously, fat is more than just TAGs. We have corrected this as suggested.

Referee #3:

In this interesting paper, Deng and colleagues use *Drosophila* to elucidate a novel pathway in which high fat diets (HFD), via a metabolite produced by the microbiota (Tyramine; TA), alter the digestive process to make it more efficient and less stress-inducing on the gut epithelium. Their experiments test and confirm many components of this signal transduction pathway to elaborate a long string of functional interactions leading from HFD to changes in the microbiota to TA production, to the TA receptor (TyrR1) to Calcium flux to CaMKII to CREB/CRTC and finally to their transcriptional targets that alter lipid digestion, lipogenesis, and mitochondria biogenesis in gut enterocytes (summarized in Fig 7e). At the end of the paper, a bit of data is offered to make a connection to insulin resistance in both flies (Fig 7) and mice (Fig 8), though the detailed pathway analysis is confined to flies. The main point of significance and interest in my opinion lies in the role of diet-induced microbiota changes and Tyramine, which is proposed as a potential dietary supplement to combat the negative effects of HFD, including metabolic syndrome, and insulin resistance in diabetes. This is a logical proposition. The study is expertly constructed and uses a wide variety of appropriate genetic tools, sensors, drugs, and assays to test this putative pathway. It is fairly well written, and logically presented, making a rather complex study easy enough to follow. There are some typos and grammatical issues, and the Discussion has a lack of focus that suggests dependence on AI editing. I liked the story overall, and all the data are

without exception consistent with the author's assertions. However, there are several links in the proposed pathway that rely either on weak (marginally significant) data, or on sets of tests that are not entirely rigorous. For these reasons (detailed below), my confidence in the overall pathway being accurate and true is not especially high. However I believe the authors should be able to bolster the weak links in their pathway with additional data that would make the study appropriate for publication.

We thank the reviewer for the positive comments. The manuscript has been significantly improved base on the reviewer's suggestions.

Major concerns:

1. The weakest apparent link in the author's proposed pathway is the role of microbiota changes, and microbiota-produced Tyramine (TA) as a regulator of gut enterocyte functions. The data relevant to this (Fig 2, 3, S4) is mostly associative, and more rigorous tests should be performed to confirm that it is actual TA from specific bacteria that is the critical secreted signaling in this system. Given the data presented (notwithstanding the requirement of TyrR1) it is possible that another bacterial metabolite executes the function in question. The authors rely heavily on TA supplementation at 1mg/ml in many experiments, and it is not determined whether this (extremely high) amount of TA has any physiological relevance. What is the concentration of TA produced by *L. Brevis* or *L. Fecalis* in the gut lumen?

One standard, elegant type of experiment that could help cement this part of the story is to

knock out TA production genes in the bacteria, and then show that the bacteria fail effect downstream gut functions, which can be rescued by TA supplementation (at physiological levels similar to those produced by gut bacteria.)

We fully understand the concerns of the reviewer. Whether bacterial-derived TA is the only or the main metabolites that regulates calcium signaling in ECs is a critical issue that needs to be addressed. We conducted two sets of experiments to answer this question. First, as suggested by the reviewer and also by reviewer 1, we successfully generated a deletion allele of *Tdc* in *L. brevis* using an HR-based strategy (**new Fig.EV4D**). Germ-free flies fed with *Tdc*^{-/-} bacteria failed to increase Ca²⁺ in ECs compared to those fed with WT bacteria (**new Fig.3E**).

Second, supplementation of the HFD with NA (Nicotinic acid) to suppress Tdc enzyme activity in TA-producing bacteria, while leaving the other species intact, also robustly suppressed the HFD-induced Ca²⁺ increase (**new Fig.EV4G**)

TA is almost insoluble in ddH₂O (1% m/v). After trying different concentrations (0.1, 0.5 or 1% w/v), we found that TA at 1mg/ml in food can mimic the effect of TA-producing bacteria on increasing Ca²⁺ signaling via TyrR/GPCR cascade (**new Fig.EV2G**).

Taken together, these results indicate that TA-producing bacteria are necessary and sufficient to promote the Ca²⁺-mediated shift in lipid metabolism.

2. The experiments with bacteria in combination with HFD are limited to Fig 3 and don't test the effects of the microbiota on downstream effects described in Figs 4-7 (e.g. on CREB & CREB targets), or in mice. Adding such experiments could increase confidence in the authors' overall conclusions.

Similar to TA (**Fig.4I**), our new results showed that flies ingested with wild type but not the *Tdc*^{-/-} *L. brevis* also displayed increased CREB activity in the gut (**Fig.4J**), further strengthening our conclusion that bacterial-derived TA regulates Ca²⁺ signaling in ECs.

3. The data in Fig 3d don't strongly associate HFD with TA-producing microbiota. Additionally, the heat maps (Fig 3d, S4) doesn't specific species, so it is impossible to tell if *L. Brevis* and *L. Fecalis*, used in later experiments, are increased by HFD.

Indeed, the original Fig3d showed that Gram-positive bacteria were enriched on HFD. The original panel in Fig.EV4C only show the Lactobacillus family was increased on HFD. We now replace it by a panel showing that L.Brevis was indeed increased on HFD. further analysis to show that L.Brevis is enriched in 16Sseq in the **new Fig.EV4C**. However, L.fecalis is actually not a commensal bacteria in fly gut. We chose it because it is a known Tdc-expressing Gram-positive bacteria, commonly found in gut microbiome of mammals. We have now clarified this in the main text in **line 235**.

4. The data in Fig 3a show a massive increase in total gut bacteria amount (CFU) from HFD. This raises the question of whether it is just more bacteria that cause the effect, and not different bacteria. Can the authors rule this out? Overall, the section about bacteria should be improved.

To answer this question, we fed the HFD animals with NA (nicotinic acid), a Tdc inhibitor, as described in point 1. In this case, the bacterial composition and abundance did not change, only the Tdc activity of the TA-producing bacteria was inhibited. As shown in **new Fig.EV4G**, NA can robustly suppress the HFD-induced Ca²⁺ increase, indicating that TA-producing bacteria rather than microbial load is responsible for Ca²⁺ signaling on HFD.

As suggested, we have also rearranged the panel of figures and re-written the section to be more appropriate and concise.

5. Many key experiments have rather high p values, in the range .05-.01 (one *). I counted 9 examples in Fig 1-7. Since low confidence results are additive in a linear pathway, this brings the probability of the whole pathway down considerably. For instance if nine results with a probability of being correct of 95% must all be correct for the pathway to be correct, then confidence in the whole pathway being correct is $.95^9=.63$. That is not very good. Hence, the authors should strive to get more data points, or more accurate data in order to lower their p values for all key experiments to $>.01$ (**). This is also true for the mouse data in Fig 8b,d.

We understand the concerns of the reviewer. We have now collected more samples for the dataset mentioned by the reviewer. The figure legends have been updated accordingly. Overall, the statistical analysis further strengthens our main conclusion in the manuscript.

6. The effect on magro gene expression is shown only for mRNA (Fig 5). Since mRNA levels do not always correlate with protein activity levels, data should be provided showing either the protein expression of this lipase, or levels of its enzymatic activity.

We fully agree with the reviewer that the enzymatic activity of magro is important for our conclusion. Magro is a major lipase secreted by ECs to digest lipids in the intestinal lumen. There are many commercial kits available to measure lipase activity, but these assays are unfeasible and misleading in our context because they would be contaminated by lipases inside ECs, such as ATGL. Instead, we have measured fecal lipids to indirectly reflect the enzymatic activity of Magro (**Fig.5F**).

7. The important lipid metabolism experiment in Fig 6d is apparently done only in the +/- CRTC condition, and not with HFD or TA administration or microbiota changes. These results should be extended and quantified. Otherwise the metabolic endpoint effects of the pathway are not so clear. Also, the sentence explaining this experiment is non-sensical and needs to be corrected (line 311-313.)

Thanks for pointing this out. We have now also examined fatty acid flux under TA conditions and found that they also more BODIPY signals in ECs while less in adipocytes (**newFig.EV7G-H**), indicating that TA also promotes locally consumption of fatty acids. Quantifications are shown in **Fig.6G-H**.

Sorry for the confusion, there were two words missing in the sentence and it has been corrected.

7. The mouse data (Fig 8) are very superficial, and don't actually demonstrate dysbiosis from HFD in mice.

We found that i.p. injection of TA also improved insulin resistance in obese mice. This suggests that supplementation with TA or TA-producing probiotics has the potential to treat diet-induced insulin resistance. However, the gut microbiome and its regulatory mechanisms are much more complicated in mice, and whether Ca²⁺ signaling to regulate lipid metabolism is conserved in mammals all awaits further study in the future. We have added a sentence about this in the discussion section (**Line 442-444**).

Minor concerns:

1. Raw metabolomics and transcriptomics data needs to be made available to readers.

The omics raw data has been uploaded to the public database. Please refer to the Data availability section for details.

2. On line 158, please note that the short time interval (4h) is meant to help show immediate effects in proximal signaling.

Yes, we agree.

3. Results from experiments should be presented in the past tense, not the present tense.

Results are logically pertinent and true only to the experiment and experimental conditions at the time the experiment was performed, in the past.

Thank you for pointing this out. We have now carefully checked the language throughout the manuscript to ensure that it is fluent and meets the requirements of the journal.

4. Not all data is presented as dot plots, but it should be. Simple bar graphs with error bars are not acceptable, since these obscure n values and data scatter.

We understand the concern of the reviewer. All data sets involving a large number of n , such as Ca^{2+} imaging, are presented as scatter plots. For those with fewer n , such as RT-q-PCR and TAG measurement, we have presented mean \pm S.E.M. with appropriate statistical analysis, and p-values have been provided for each plot. We are happy to re-analyze all these data in scatterplot form if required.

5. Fig 4a, b are missing the control conditions, with Gaq-Ri and TyrRa-Ri alone (without TA addition). These should be included.

The controls are now included in **new Fig.4A-B**.

6. Line 313 mentions "data not shown", which needs to be shown.

Food intake by the CAFÉ assay is now shown in **Fig. EV7F**. The methods have been updated accordingly.

Dear Hansong,

Thank you for submitting your revised manuscript to The EMBO Journal. Your manuscript has now been seen by the two original reviewers, and you can find their comments below.

As you can see, while reviewer #1 finds that their concerns have been sufficiently addressed, reviewer #2 indicates that several of their initially raised points need further experimental support before they can support publication of the study. Based on this input, I would like to invite you to address the remaining comments by reviewer #2 and the following editorial issues:

1. Please check that the funding information is correct and identical both in the manuscript and our online system. Currently, Tongji University Basic Scientific Research-Interdisciplinary Fund [grant no. 2000123424] is missing from our online system. Please also include funding information in the Acknowledgements section.
2. Please make sure that the order of the sections in the manuscript is as follows: abstract, introduction, results, discussion, materials & methods, data availability section, acknowledgments, disclosure statement and competing interests, references, main figure legends, tables, expanded figure legends.
3. Please rename "Conflict of interest" section into "Disclosure and competing interests statement" (further info: <https://www.embopress.org/page/journal/14602075/authorguide#conflictsofinterest>). Please also remove the duplicate sections at the title pages of the main manuscript and Appendix.
4. CRedit has replaced the traditional author contributions section because it offers a systematic, machine-readable author contributions format that allows for more effective research assessment. Please remove the Authors Contributions from the manuscript and use the free text boxes beneath each contributing author's name in our online submission system to add specific details on the author's contribution. More information is available in our guide to authors
5. We can accommodate up to five EV figures that are collapsible/expandable online, therefore please consider if you would like to select five of the figures for this format. EV Figures should be cited as 'Figure EV1, Figure EV2' etc. in the text. We would also need individual production quality figure files in the .eps, .tif, or .jpg format (one file per figure). Further information on the format is available here: <https://www.embopress.org/page/journal/14602075/authorguide#expandedview>.
6. The remaining figures and their legends should be renamed Appendix Figure S1 etc. with the corresponding callouts and legends placed below the figures; duplicate figure labels should be removed from Appendix; nomenclature should be updated in the Table of contents in Appendix file, and page numbers should be added.
7. Please remove movie legends from the manuscript and zip with each corresponding movie file.
8. Please submit source data as requested by our data editor after invitation to revise the manuscript. I have attached the checklist below - please note that the figure panel numbers refer to the previous version of the manuscript.
9. Our data editors have flagged the following issues in figure legends that need correcting:
 - Please note that the legends for figures EV3d, f and EV5g are missing in the manuscript. This needs to be rectified.
 - The legend for figure EV3e is incorrectly labelled as EV3d, please correct.
 - The legend for figure EV5h is incorrectly labelled as EV5g, please correct.
 - Please define the annotated p values **/* in the legend of figure 4j; 5d; EV 1a; as appropriate.
 - Please indicate the statistical test used for data analysis in the legends of figures 4j; 6a-c; EV1c-d; EV2e-f; EV7d.
 - Please note that in figures 4d, f-g, i, m; 5c; 6h; 7a-c; EV2g; EV3a; EV4g; there is a mismatch between the annotated p values in the figure legend and the annotated p values in the figure file that should be corrected.
 - Please define the box plots in terms of minima, maxima, centre, bounds of box and whiskers, and percentile in the legends of figures EV4a-b.
 - Please describe the number and nature of the replicates in the legends of figures 2e; 5d; 6c; EV5c, e; EV6c.
 - Please describe the nature of replicates in the legends of figures 4l-n; 5e, g; EV1f, h-i; EV2a, f; EV4a-b; EV5d; EV7f.
 - Please define the error bars in the legends of figures EV2c; EV5a.
 - Please define the scale bar for figures EV1e; EV7b.
 - Please define the white arrowheads in the legend of figure EV8b.
10. Papers published in The EMBO Journal are accompanied online by a 'Synopsis' to enhance discoverability of the manuscript. Please submit synopsis text that consists of A) a short (1-2 sentences) summary of the findings and their significance, B) 3-4 bullet points highlighting key results and C) a synopsis image that is 550x300-600 pixels large (width x height, jpeg or png format). You can either show a model or key data in the synopsis image. Please note that the image size is rather small, and that text needs to be readable at the final size.

Please feel free to contact me if you have any questions regarding this final revision. Please use the link below to upload the revised files.

Thank you for the opportunity to consider your work for publication, and I look forward to receiving your revised manuscript.

With best wishes,

leva

We realize that it is difficult to revise to a specific deadline. In the interest of protecting the conceptual advance provided by the work, we recommend a revision within 3 months (5th Aug 2024). Please discuss the revision progress ahead of this time with the editor if you require more time to complete the revisions.

Referee #1:

I thank the reviewers for their efforts to improve the manuscript. I am satisfied with the revision, and I recommend the manuscript for publication.

Referee #2:

Review of Ma et al.

The authors have addressed some of the points raised in the previous review. However, several issues still remain. They are listed below:

1) An important conclusion of this manuscript is that TA produced by gut bacteria signals a response towards HFD. It is therefore critical that the authors say clearly what if any is the role of TA produced by the host in HFD. In their revised manuscript the authors have taken an RNAi line for TDC2, expressed it in the gut and looked at cytoCa²⁺ (Fig EV3A). There are two problems with this experiment. First it has not been performed on HFD. Second TA could be synthesized anywhere else in the body and reach the gut through the haemolymph. The authors should drive the RNAi using a ubiquitous GAL4 and in animals that lack gut bacteria and are on HFD. Appropriate genetic controls are also essential.

2) The explanation for excess TAG secretion after reduced Magro expression is not convincing. What is the enzymatic step catalysed by Magro? Why are more TAGs excreted when Magro is downregulated.

3) The authors state that mitochondrial biogenesis goes up on HFD. However, their measurements are primarily on CRTCOE flies. The only direct measurement of mitochondria on HFD are EM images (Fig EV7C) which to me appear no different between the two conditions. Another direct measurement of mitochondrial biogenesis on HFD should be included.

Apart from the above there are some minor points that need addressing:

1) What is the manipulation in Figure 6I?

2) In the response to reviewers the author mention that 73% of DEG contain a CRE motif. This should be mentioned in the text/figures and the analytical method mentioned in the methods.

3) The time scale at which lipid uptake occurs vs the change in [Ca²⁺] upon incubation with fatty acids should be mentioned in the text at an appropriate place.

4) Fig 6A mentions 4 common pathways between IP3ROE and CRTCOE. Strangely these pathways are missing from the top 20 KEGG enrichment pathways shown in Figure 6B. The authors should clarify this point in the main text.

5) The blot in Fig EV7A should be quantified.

6) BODIPY and LIPIIDTox levels should be quantified in the adipocytes (Fig EV7H).

Referee #1:

I thank the reviewers for their efforts to improve the manuscript. I am satisfied with the revision, and I recommend the manuscript for publication.

Referee #2:

Review of Ma et al.

The authors have addressed some of the points raised in the previous review. However, several issues still remain. They are listed below:

1) An important conclusion of this manuscript is that TA produced by gut bacteria signals a response towards HFD. It is therefore critical that the authors say clearly what if any is the role of TA produced by the host in HFD. In their revised manuscript the authors have taken an RNAi line for *TDC2*, expressed it in the gut and looked at cytoCa²⁺ (Fig EV3A). There are two problems with this experiment. First it has not been performed on HFD. Second TA could be synthesized anywhere else in the body and reach the gut through the haemolymph. The authors should drive the RNAi using a ubiquitous GAL4 and in animals that lack gut bacteria and are on HFD. Appropriate genetic controls are also essential.

Our main conclusion is that TA-expressing bacteria were enriched in HFD and led to Ca²⁺ activation in ECs to regulate lipid metabolism.

As mentioned by the reviewer, in the previous version, we showed that knockdown of peripheral *Tdc1* in ECs didn't affect HFD-mediated EC [Ca²⁺] levels, and that neuronal *Tdc2* expression was largely unchanged under HFD conditions (Fig.EV3A-B).

As suggested by the reviewer, we have performed more experiments to rule out the role of host-derived TA. As shown in new Fig.EV3C, ubiquitous knockdown of *Tdc2* did not affect intestinal lipid levels under both ND and HFD conditions, further confirming that TA derived from intestinal microbiota is responsible for Ca²⁺/CREB-mediated lipid metabolism. The main text has been updated accordingly (lines 212-214).

2) The explanation for excess TAG secretion after reduced *Magro* expression is not convincing. What is the enzymatic step catalyzed by *Magro*? Why are more TAGs excreted when *Magro* is downregulated.

Dietary fats (mainly TAGs) must be digested by lipase into free fatty acids or MAGs before they can be absorbed by ECs. When lipase was inhibited, more fats were excreted in the feces. For example, Orlistat, an FDA-approved anti-obesity drug that inhibits PTL (pancreatic lipase), often leads to steatorrhea.

Magro is the functional counterpart of PTL in *Drosophila* (PMC8257970), which is highly expressed in ECs and secreted into the intestinal lumen to digest dietary lipids. TAG excretion (fecal lipids) was used as an indirect measure of *Magro* activity (more fecal lipids means less absorption and less lipase activity).

To further address the reviewer's concerns, we now directly measured its activity in the gut lumen using a lipase activity kit (the details are updated in the Methods section). As shown in new Fig.5G, lipase activity in the gut lumen was markedly reduced under HFD or by CRTC overexpression, further confirming that *Magro* is transcriptionally repressed by the Ca²⁺ signaling cascade to suppress dietary lipid digestion. The main text has been updated accordingly (lines 321-322).

Please note the fecal ORO staining in original Fig.5F was moved to Appendix Figure S1D to conform to the journal's guideline (5 EV figures maximum).

3) The authors state that mitochondrial biogenesis goes up on HFD. However, their measurements are primarily on CRTCOE flies. The only direct measurement of mitochondria on HFD are EM images (Fig EV7C) which to me appear no different between the two conditions. Another direct measurement of mitochondrial biogenesis on HFD should be included.

Thanks for the suggestion. To quantify the mitochondrial content in fly guts under ND or HFD, WB against ATP5A was performed. As shown in new Appendix Figure S2E, a significant increase in mitochondrial content was indeed observed under HFD conditions.

Please note that Fig. EV7 has been rearranged to fit the new data and renamed Appendix Figure S2 to conform to the journal's guideline.

Apart from the above there are some minor points that need addressing:

1) What is the manipulation in Figure 6I?

TAG levels in adipocytes were measured in Fig.6I when fed diets supplemented with or without tyramine (TA, 1 mg/ml). Figure legend has been updated accordingly.

2) In the response to reviewers the author mention that 73% of DEG contain a CRE motif. This should be mentioned in the text/figures and the analytical method mentioned in the methods.

We apologize for that. We now included it in new Appendix Figure S1C and in the main text (lines 306-309), and the methods sections has been updated as well.

3) The time scale at which lipid uptake occurs vs the change in [Ca²⁺] upon incubation with fatty acids should be mentioned in the text at an appropriate place.

We have mentioned in line 166-175 by saying that "... although animals fed with a FFA mixture derived from coconut oil for 4h show elevated enteric lipids, cytoCa²⁺ in ECs is largely unchanged". We believe it can at least partially address the reviewer's concerns.

4) Fig 6A mentions 4 common pathways between IP3ROE and CRTCOE. Strangely these pathways are missing from the top 20 KEGG enrichment pathways shown in Figure 6B. The authors should clarify this point in the main text.

Sorry for the confusion. The four pathways mentioned (glutathione metabolism, proteostasis..) are known processes regulated by the Ca²⁺/CRTC cascade that we have described in the previous study(PUBMDId 35933423), that's why we denoted them in the Venn diagram in Fig.EV6A(now Appendix Figure S1A). The Top 20 KEGG pathway of these DEGs were shown in Fig.EV6B (Appendix Figure S1B). We now have clarified it in the figure legend as well as in main text (line 299-300).

5) The blot in Fig EV7A should be quantified.

Quantification was shown in Fig.6C in the previous version.

6) BODIPY and LIPIdTox evels should be quantified in the adipocytes (Fig EV7H).

The quantifications are shown in new Appendix Figure S2H and J. Again, please note that the figure has been rearranged and renamed to conform to the journal's guidelines.

We greatly appreciate the valuable time of the reviewers. Their comments have greatly improved the quality of the manuscript.

Dear Hansong,

Thank you for addressing most of the remaining points. I sincerely apologise for the slow process from our side due to the high number of submissions that we experience at the moment. Upon checking your manuscript, I am afraid that I noticed a few remaining issues that need to be fixed before I can formally accept it for publication:

1) For several figure panels, conflicting information is provided on the nature of replicates, please check and clarify:

Fig EV1 F - "Biological triplicates were performed. $n \geq 10$.";

EV1 H-I - "n=6 for each condition. Mean {plus minus} S.E.M were shown for three independent replicates";

EV2 A - "n=6. Mean{plus minus} S.E.M for three independent replicates.";

EV5 D - "Biological triplicates were performed. $n=10$.";

Appendix Figure S2 G - "Biological triplicates were performed 10 animals per group".

2) In the legend for figure EV4 A-B, it is stated that the statistical analysis is performed on duplicates. This is not permitted in our guidelines. Instead, please show the values of each of the duplicates as separate data points. If more data points were used than stated, please correct this and indicate what values are indicated by the centre of the box (usually mean or median), the outline of the box and the whiskers, as also previously requested by our data editors.

3) In the source data for Figure 1K, columns H and I, values derived from each ND w118 gut and the HFD w118 guts 1-3 have identical three decimal numbers after the comma (please see attached file). Since this appears statistically highly unlikely, an explanation on how these values were obtained would be very helpful.

4) I would like to propose some minor edits in the manuscript title, abstract and synopsis (please see below and the attached manuscript text file). I have also written a short blurb that will accompany the title of your manuscript in our online system. Please let me know if any corrections or adjustments are needed:

Title:

Gut microbiota metabolite tyramine ameliorates high-fat diet-induced insulin resistance via increased Ca^{2+} signaling

Blurb:

Dietary fats enhance prevalence of tyramine-producing bacteria, leading to suppression of lipid accumulation in gut enterocytes.

Synopsis:

High-fat diet (HFD)-induced alterations in gut microbiota, also known as dysbiosis, can alter host physiology via metabolite production. This study reveals a conserved role of the gut microbiota-derived bioamine tyramine in suppression of HFD-evoked insulin resistance.

- HFD increases prevalence of tyramine-producing bacteria in *Drosophila*.

- Tyramine activates cytosolic Ca^{2+} signaling in enterocytes via its receptor TyR1.

- Tyramine-TyR1-dependent cytosolic Ca^{2+} changes activate the CRTC/CREB complex to suppress dietary lipid digestion and promote mitochondrial biogenesis.

- Tyramine suppresses insulin resistance in HFD-fed fruit flies and mice.

Thank you again for giving us the chance to consider your manuscript for The EMBO Journal. I look forward to receiving your input on these final points.

With best regards,

leva

leva Gailite, PhD
Senior Scientific Editor
The EMBO Journal
Meyerhofstrasse 1
D-69117 Heidelberg
Tel: +4962218891309
i.gailite@embojournal.org

We realize that it is difficult to revise to a specific deadline. In the interest of protecting the conceptual advance provided by the work, we recommend a revision within 3 months (17th Sep 2024). Please discuss the revision progress ahead of this time with the editor if you require more time to complete the revisions.

The authors addressed the minor editorial issues.

Dear Hansong,

Thank you for addressing the final points and for your clarifications on replicates used and the source data values. I am now pleased to inform you that your manuscript has been accepted for publication. Congratulations on a nice study!

Finally, we would like to promote your manuscript among the Chinese readership. Therefore, we would like to invite you to prepare a short summary of the manuscript in Chinese (1500-2000 Chinese characters), which we will promote on the WeChat platform 'BioArt' with more than 610,000 followers.

If you are interested in this opportunity, we recommend covering the article very close to its online publication date. Thus, ideally we would appreciate if you could send us a draft within the next 7 working days. Please let us know whether or not you would be interested in contributing such a short summary in Chinese.

I have included below some general guidelines on how to prepare a summary and a link to recent examples for your reference. Please let me know if you have any questions about this.

If you have any questions, please do not hesitate to contact the Editorial Office. Thank you again for this contribution to The EMBO Journal and congratulations on a successful publication!

Best regards,

Ieva

General WeChat Summary Guidelines

1. These summary articles are meant to be targeting general audience so please limit the use of specialized technical terms, acronyms and jargon.
2. A summary usually starts with brief background information of the reported work, which is followed by explaining the findings in some detail, and ends with a short review of the conclusions as well as the implications of the work and future directions for the research.
3. The summary should at least contain one graphical item, such as a scheme or a figure from the paper.
4. Please provide ONE SINGLE document containing all text and graphical materials, ideally as a Word.docx or .doc file. Please DO NOT provide the document as a .pdf file.
5. Please DO NOT publicly release the document before the paper is officially published online.

Summary Examples

EMBO J | 罗招庆/欧阳松应揭示谷酰胺脱氨酶MvcA的去泛素化功能
